# Client proximity enhancement inside cellular membrane-less compartments governed by client-compartment interactions

Daesun Song [1], Yongsang Jo[1], Jeong-Mo Choi [2,3] & Yongwon Jung [1✉]

Membrane-less organelles or compartments are considered to be dynamic reaction centers for spatiotemporal control of diverse cellular processes in eukaryotic cells. Although their formation mechanisms have been steadily elucidated via the classical concept of liquid–liquid phase separation, biomolecular behaviors such as protein interactions inside these liquid compartments have been largely unexplored. Here we report quantitative measurements of changes in protein interactions for the proteins recruited into membrane-less compartments (termed client proteins) in living cells. Under a wide range of phase separation conditions, protein interaction signals were vastly increased only inside compartments, indicating greatly enhanced proximity between recruited client proteins. By employing an in vitro phase separation model, we discovered that the operational proximity of clients (measured from client–client interactions) could be over 16 times higher than the expected proximity from actual client concentrations inside compartments. We propose that two aspects should be considered when explaining client proximity enhancement by phase separation compartmentalization: (1) clients are selectively recruited into compartments, leading to concentration enrichment, and more importantly, (2) recruited clients are further localized around compartment-forming scaffold protein networks, which results in even higher client proximity.

[1] Department of Chemistry, Korea Advanced Institute of Science and Technology, Daejeon 34141, Republic of Korea. [2] Natural Science Research Institute, Korea Advanced Institute of Science and Technology, Daejeon 34141, Republic of Korea. [3] Department of Chemistry, Pusan National University, Busan 46241, Republic of Korea. ✉email: ywjung@kaist.ac.kr

Eukaryotic cells utilize various interior compartments to control highly complex biomolecular reactions in space and time. In addition to conventional membrane-bound organelles such as the endoplasmic reticulum or Golgi, many membrane-less compartments, which are condensed with distinct sets of biomolecules without discrete lipid bilayer barriers (therefore also termed biomolecular condensates), have been reported[1]. Examples of these membrane-less organelles include stress granules, p-bodies, and nucleoli, which are known as essential hubs of cellular processes such as signal transduction, stress response, and gene expression[2]. Many membrane-less compartments show remarkable liquid-like properties such as high inner diffusivity, reversible formation, and free (yet possibly controlled) exchange of molecules with their surroundings[3]. Recent studies indicate that repeated folded protein domains[4] or intrinsically disordered proteins/regions (IDPs/IDRs)[5–7] can drive liquid–liquid phase separation (LLPS), and that this is the major formation principle of compartmentalized biomolecular condensates (also termed droplets)[2,8,9]. Among numerous components of membrane-less organelles, a small number of IDPs (sometimes even a single IDP) have been shown to be necessary and sufficient to form condensates, both in vitro and in cells[1–3], and are termed scaffolds. Other components are rather passively recruited into the condensates and hence called clients.

Although more information on the acting mechanisms of biomolecular LLPS is continuously being revealed, our understanding of biomolecular behaviors such as protein interactions and enzymatic reactions inside membrane-less compartments is still very limited. Several studies reported enhanced biomolecular reactions by increased reactant concentrations inside synthetic model compartments[10–16]. For example, ribozyme cleavage was enhanced 70-fold by RNA enrichment (~3000-fold concentration increase) in the dextran-rich phase of a polyethylene glycol (PEG)/dextran aqueous two-phase system[10]. Multiple studies have also demonstrated enhanced ribozyme catalysis and RNA polymerization reactions inside compartments formed by complex coacervation between various pairs of cationic and anionic polymers (e.g., carboxymethyl dextran/poly-lysine and poly-dia-llyldimethyl-ammonium/RNA)[12–14]. An RNA deadenylation rate was also enhanced in in vitro compartments formed by the IDRs of two interacting translation-regulating proteins and RNA[16]. Recently, a few studies have also described distinct biomolecule activities inside compartments formed by more physiologically relevant protein LLPS; actin polymerization was locally accelerated by recruitment of an actin nucleation factor to multi-domain nephrin-Nck-N-WASP protein condensates[4,17]. The unstructured protein tau also formed a liquid condensate, which recruited and concentrated tubulin; the enriched tubulins were polymerized to microtubule bundles in tau droplets[18]. In the other study, double-stranded DNA was shown to be destabilized inside protein compartments composed of IDP DDX4[19].

Natural membrane-less organelles often contain more than a hundred different components, among which a few are condensate-forming scaffold proteins and others are recruited as clients. Recent studies indicated that specific partitioning of clients and scaffolds in condensates can be influenced by various factors such as crowding environments[20] and condensate compositions[21]. Dynamic mRNA partitioning to cellular stress granules or to processing bodies was also reported[22], and a theoretical model was used to study how client–scaffold interactions govern condensate stability[23]. To fully elucidate the working principles of these compartments as temporal reaction centers in cells, it is important to understand how selectively enriched clients differently react inside protein compartments compared to outside. Here we report the real-time quantitative measurements of enhanced interactions between client molecules upon recruitment into membrane-less IDR compartments in living cells. Various cellular protein condensates were formed with tandemly repeated IDRs or opto-controllable IDRs. For subsequent control of client recruitment into condensates, different IDRs or IDR fragments were fused to clients, in order that IDR-fused clients can be recruited to IDR condensates via IDR–IDR interactions.

To quantitatively monitor protein interactions inside condensates, two fluorescent protein (FP) probes, which can generate fluorescent signals by FP self-interactions, were used as clients. By employing IDR-based condensates and clients, as well as interaction-responsive FP probes, we systematically investigated FP interaction dynamics inside and outside of compartments formed in living cells. We observed unexpectedly high and rapid enhancement of client–protein interactions when recruited into the condensates, and this effect cannot be explained solely by the increased inner concentrations of client proteins by enrichment. In vitro experiments indicated that the acting proximity of clients inside condensates is significantly higher than the expected proximity based on actual inner client concentrations. In addition, these proximity increases are augmented for stronger scaffold–client IDP interactions. This implies that in addition to the well-known protein enrichment effect by recruitment, there is an additional proximity enhancement effect due to local entrapment of clients around scaffold protein networks inside compartments.

## Results

**Fluorescence complementation (FC) of scaffold-fused probes inside repeated-IDR compartments.** Following the natural systems, we used various IDRs not only for cellular membrane-less compartment formation but also for client recruitment into these compartments. We employed the disordered N-terminal prion-like domain (residues 1–214) of the FUS (Fused in Sarcoma) protein. FUS is a well-characterized condensate-forming IDP and multiple studies reported recruitment of various IDPs into FUS condensates[6,24,25]. To vary the degree of cellular compartment formation and also to search for ideal FUS fragments for client recruitment, we constructed a series of FUS variants with different lengths (Fig. 1a). Previous studies indicated that LLPS could be strongly enhanced by clustering or tandemly repeating IDR proteins, likely by having more interactable residues on scaffolds[24,26,27]. The degree of phase separation (the number/size of observable droplets) clearly increases as the protein length increases in vitro, although truncated FUS variants show almost no LLPS (Fig. 1a).

Cellular membrane-less compartment formation of these FUS proteins was examined by expressing mCherry (mCh)-fused FUS variants in cells. Consistent with the in vitro protein solution test, more protein condensates (red puncta) were observed with longer FUS proteins in diverse cell lines (Fig. 1a and Supplementary Fig. 1). Most FUS condensates were observed in the nucleus. More than 60% of tandemly repeated 2.0 FUS phase separated into droplets (puncta sizes > 0.01 μm$^2$), while less than 10% of wild-type 1.0 FUS formed droplets (Fig. 1b). Protein condensates were hardly observed with truncated FUS proteins such as 0.5 N, 0.5 C, and 0.75 FUS. To investigate the dynamics inside mCh-FUS condensates, we conducted a fluorescence recovery after photobleaching (FRAP) analysis (Fig. 1c). Relatively fast signal recoveries of FUS droplets ($t_{1/2}$ ~ 50 s) indicate dynamic protein diffusivity and exchange with surroundings, which are representative properties of liquid-like membrane-less compartments.

To quantitatively monitor protein interactions inside FUS compartments in cells, we first directly fused an interaction-responsive FP probe to scaffold FUS proteins. The previously

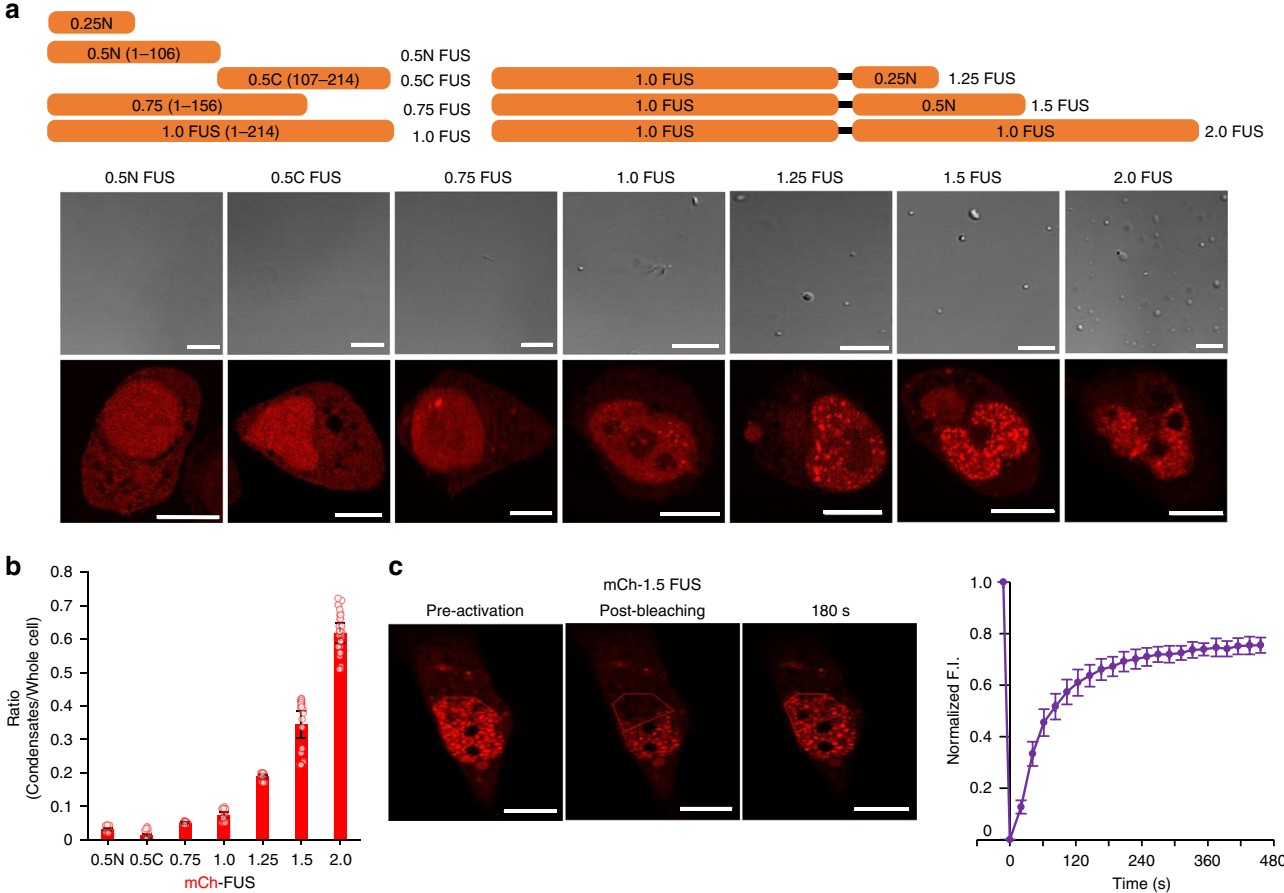

**Fig. 1 Cellular compartment formation with length-varied FUS. a** Protein condensate formation of mCherry (mCh)-fused FUS constructs with varied lengths in HeLa cells (bottom) or length-varied FUS in a solution (middle). Schematic diagrams of length-varied FUS constructs (sequences in Supplementary Information) are shown (top). **b** Total mCh intensities of condensates (puncta sizes > 0.01 μm²) divided by whole cell mCh intensities. Data are presented as mean values with ±1 SD as error bars (n = 32 for 0.5 N, 0.5 C, 0.75, n = 31 for 1.0, 1.25, n = 33 for 1.5, 2.0 FUS; n: biologically independent cells examined over three independent experiments). **c** FRAP recovery images and profiles of scaffold proteins of mCh-1.5 FUS condensates. Data (point) are presented as mean values with ±1 SD as error bars (n = 33 cells from three independent experiments). Scale bars: 10 μm.

developed homo-molecular fluorescence complementation (Homo-FC) probe[28] was employed as a FP probe. FC is induced by close proximity between probes, and FC-induced (turn-on) signal generation is well-suited for quantitative cellular visualization. Moreover, compared to conventional two-component split FC protein probes, the Homo-FC probe presents low signals in the absence of proper proximity enhancement, and only single construct expression is required[28]. The green fluorescent protein (GFP)-based Homo-FC probe was fused to length-varied mCh-FUS constructs and again expressed in HeLa cells (Fig. 2a). Consistent with mCh-FUS construct experiments, Homo-FC constructs with longer FUS proteins formed more cellular condensates (Fig. 2b), even though slightly fewer proteins were found in droplets (Fig. 2c vs. 1b). Importantly, as the FUS length increased, complementation green signals of Homo-FC were also significantly amplified (Fig. 2b). The complementation signal (GFP)-to-mCh ratio of 2.0 FUS–Homo-FC was 36-fold higher than that of 0.5 N FUS–Homo-FC (Fig. 2d). Stronger interactions between longer FUS proteins might bring fused Homo-FC probes close together for complementation. In addition, it is possible that condensate formation and subsequent probe enrichment inside contribute to enhanced complementation of longer FUS–Homo-FC constructs. Condensate partition coefficients (PCs; mCh signal ratios between inside/outside of condensates) show that protein enrichment also increased as the FUS length increased (Fig. 2e). Enhanced complementation of Homo-FC inside long

FUS condensates was also observed in other cell lines (Supplementary Fig. 2a). FRAP assays again showed dynamic diffusivities of Homo-FC-fused FUS condensates (Supplementary Fig. 2b).

**FC of recruited client probes inside repeated-IDP compartments.** To better investigate natural compartments consisting of scaffolds and clients, we next designed the Homo-FC probe as a client that can be recruited into FUS condensates. The half-truncated 0.5 N FUS was fused to the client Homo-FC; although 0.5 N FUS is too short to form droplets as a scaffold by itself (Fig. 1a) or enhance Homo-FC probe complementation (Fig. 2b), the construct can be recruited into condensates made of longer FUS variants, presumably via interactions between FUS fragments. To examine client recruitment, 0.5 N FUS was first fused to GFP (0.5 N FUS-GFP) and co-expressed with mCh-2.0 FUS, which shows the strongest tendency for LLPS among the tested variants (Fig. 3a). After 18 h transfection, 0.5 N FUS-GFP signals were clearly higher inside mCh-2.0 FUS condensates (Fig. 3b). The PC value of 0.5 N FUS-GFP was 2.1, whereas the mCh-2.0 FUS scaffold PC was 3.1 (Fig. 3c). Similar 0.5 N FUS-GFP recruitment was observed in the condensates made of mCh-1.5 FUS (Supplementary Fig. 3). It is noteworthy that relative expression levels of GFP clients and mCh scaffold proteins were consistent across different cells in our co-expression system, even when the overall protein expression levels widely varied (~10-fold) (Fig. 3d). In addition, similar recruitment

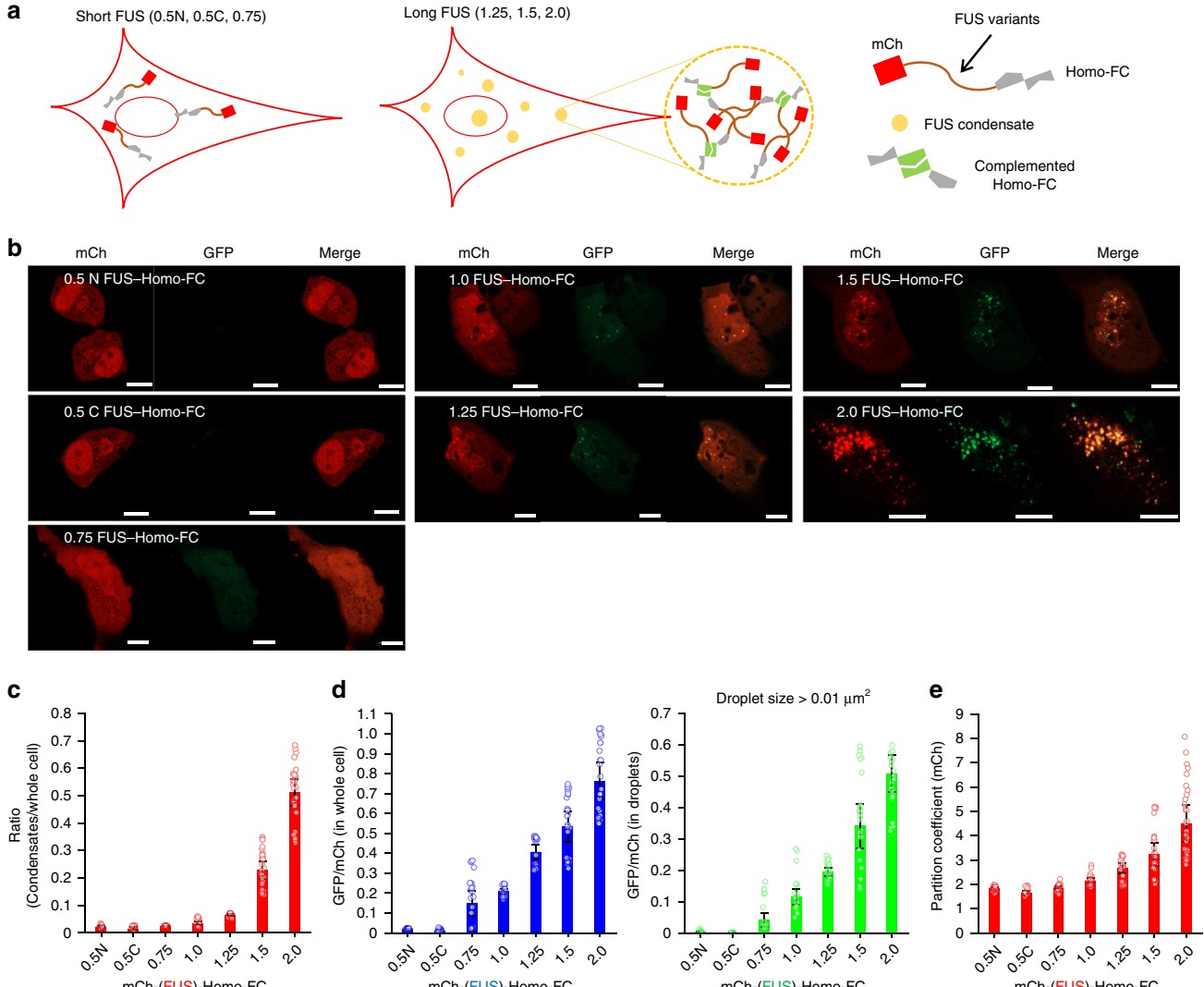

**Fig. 2 Fluorescence complementation of scaffold-fused probes inside repeated-FUS compartments. a** Schematic illustration of cellular condensate formation of long FUS constructs and FUS-fused Homo-FC probe complementation (green) inside FUS condensates. **b** Fluorescence (mCh and GFP) images of cells expressing mCh-FUS–Homo-FC proteins with varied FUS. Scale bars: 10 μm. **c** Total mCh intensities of condensates (puncta sizes > 0.01 μm$^2$) divided by whole cell mCh intensities for length-varied mCh-FUS–Homo-FC constructs. Data are presented as mean values with ±1 SD as error bars ($n = 33$ cells from three independent experiments). **d** Total GFP (complemented Homo-FC)-to-mCh ratios in whole cells (left) or inside condensate droplets (right). Data are presented as mean values with ±1 SD as error bars ($n = 33$ cells from three independent experiments). **e** mCh-FUS partition coefficients (PCs; mCh ratios inside/outside condensates). Although partition coefficients of all FUS variants are shown, condensates of short FUS variants (0.5 N, 0.5 C, 0.75 FUS) were rarely observed. Data are presented as mean values with ±1 SD as error bars ($n = 33$ cells from three independent experiments).

of 0.5 N FUS into 2.0 FUS condensates was observed when fused FPs were switched (i.e., mCh client + GFP scaffolds) (Supplementary Fig. 4).

To monitor complementation of recruited Homo-FC probes inside FUS condensates, 0.5 N FUS was next fused to the Homo-FC probe (0.5 N FUS–Homo-FC) and co-expressed with mCh-fused 1.5 FUS or 2.0 FUS scaffolds (Fig. 3e). Strong FC signals were observed inside FUS condensates (Fig. 3f and Supplementary Fig. 5). The data indicate that proximity-induced complementation between Homo-FC probes was greatly enhanced by recruitment into liquid-like FUS condensates. GFP/mCh signal ratios of client 0.5 N FUS–Homo-FC were nearly identical to those of scaffold-fused Homo-FC (Fig. 2d vs. 3g). Overall, FC signals were higher with 2.0 FUS condensates than the 1.5 FUS condensates, possibly due to a slightly higher FUS density or stronger 0.5 FUS recruitment of the 2.0 FUS scaffold.

Interestingly, observed complementation enhancement inside condensates cannot be simply explained by the increased concentration due to recruitment. Here, 0.5 N FUS–Homo-FC concentration was not directly measured since it requires additional FP (e.g., Blue FP) labeling (and three-color imaging), which might perturb accurate FC signal measurements by FP cross-talks. Therefore, we assumed that 0.5 N FUS–Homo-FC enrichment inside 2.0 FUS condensates is similar to that of 0.5 N FUS-GFP, which was only ca. twofold (Fig. 3c). We examined cells with a wide range of protein expression levels; the difference between the lowest and highest client expression levels is ~5-fold. Therefore, it is expected that the cytosol of cells (without condensates) with the highest expression levels may contain similar or even greater amount of client proteins than the condensates formed in cells with the lowest expression levels have. However, even cells with the lowest expression levels show

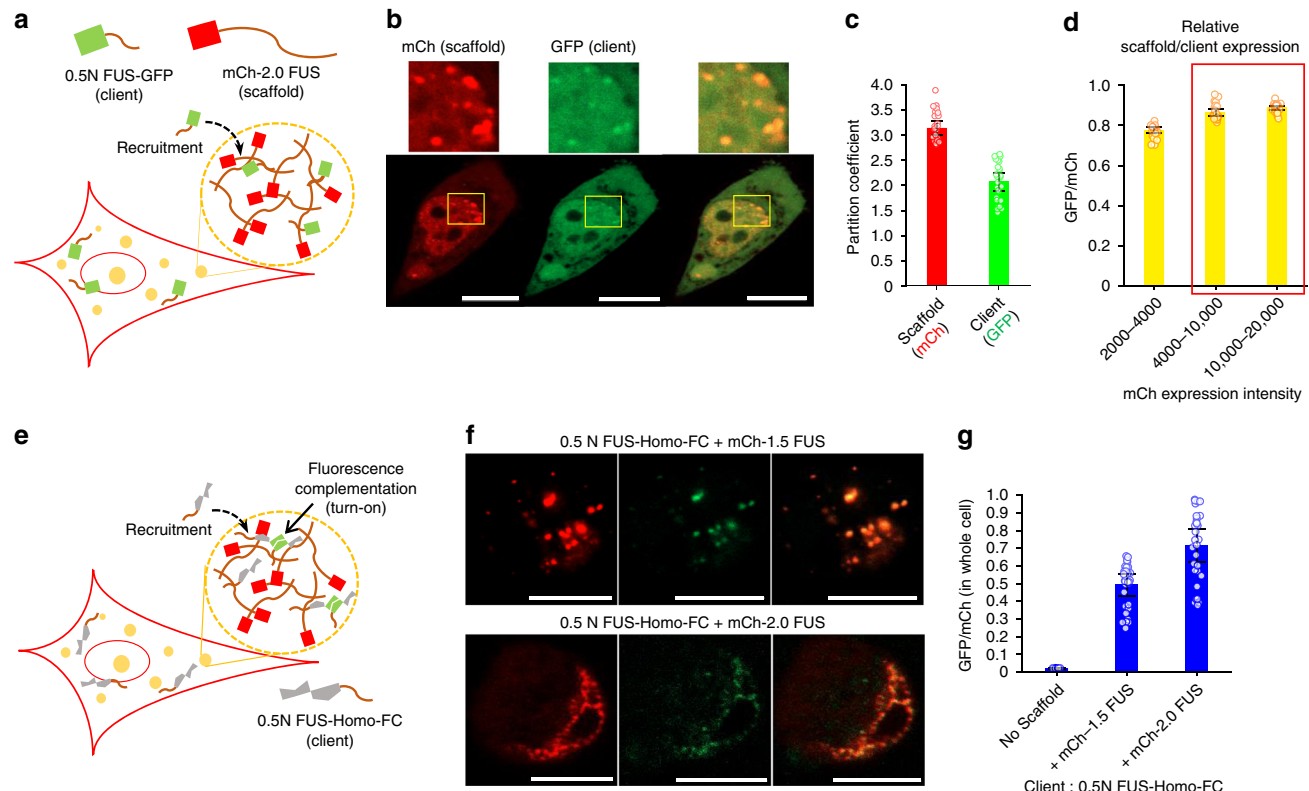

**Fig. 3 Fluorescence complementation of recruited client probes inside repeated-IDP compartments. a** Schematic illustration of cellular condensate formation of mCh-2.0 FUS and 0.5 N FUS-GFP recruitment. **b** Fluorescence (mCh and GFP) images of cells expressing mCh-2.0 FUS (scaffold) and 0.5 N FUS-GFP (client). **c** Partition coefficients of scaffold (mCh) and client (GFP) between inside and outside condensates. Data are presented as mean values with ±1 SD as error bars (n = 33 cells from three independent experiments). **d** Relative expression levels of 0.5 N FUS-GFP and mCh-2.0 FUS (GFP/mCh). Expression levels are categorized based on mCh expression intensities. Cells with mCh expression intensities between 4000 and 20,000 (red box) were used for data analysis. Data are presented as mean values with ±1 SD as error bars (n = 30 cells from three independent experiments). **e** Schematic illustration of cellular condensate formation of mCh-2.0 FUS, 0.5 N FUS–Homo-FC recruitment, and client complementation for (green) fluorescence turn-on. **f** Fluorescence (mCh and GFP) images of cells expressing mCh-1.5 FUS or mCh-2.0 FUS (scaffold) and 0.5 N FUS–Homo-FC (client). **g** Total GFP (complemented Homo-FC)-to-mCh ratios for cells without (no scaffold) or with 1.5 FUS and 2.0 FUS condensates. Data are presented as mean values with ±1 SD as error bars (n = 33 cells from three independent experiments). Scale bars: 10 μm.

strong complementation signals inside FUS condensates, while the signals were very low with the client alone (without condensates) in cells with the highest expression levels (Supplementary Fig. 6). The data suggest that there is an additional factor to enhance the proximity of client probes inside condensates, in addition to the simple client concentration enrichment.

**Real-time observation of FC inside light-induced IDP compartments.** Although we directly observed enhanced FP complementation inside cellular IDR compartments, analyzed images were obtained from accumulated LLPS processes and complementation signals over 18 h during transfection. It is possible that the material properties of the droplets change (known as aging of droplets[6,25]) during this period. To reduce the potential aging effect and observe more immediate dynamics of client–scaffold interactions, we turned to light-induced LLPS systems. Recently, light-induced protein clustering was successfully used to temporally trigger and analyze IDR phase separation behaviors in living cells[29–31]. We used a light-activatable protein CRY2, which undergoes oligomerization in response to a 488 nm light[32,33]. Previously, CRY2 and CRY2olig (a mutated version (E490G) of CRY2 with a higher clustering ability) were successfully applied to the FUS system to induce rapid LLPS in cells[29]. Similarly, CRY2 and CRY2olig were fused to mCh-1.0 FUS, and the resulting light-activatable 1.0 FUS constructs were expressed

in HeLa cells (18 h transfection). Upon 488 nm laser illumination only for 10 s, strong LLPS was observed for both constructs (Supplementary Fig. 7). We chose CRY2olig for subsequent client recruitment experiments, as it provided more consistent condensate formation regardless of protein expression levels (Supplementary Fig. 7).

The 0.5 N FUS–Homo-FC client was co-expressed with mCh-CRY2olig-1.0 FUS (Fig. 4a). Multiple FUS condensates were rapidly formed upon 10 s light illumination and the illuminated cells were imaged alive for the next 4 min to monitor client recruitment and complementation. Weak, yet noticeable, complementation signals of 0.5 N FUS–Homo-FC were observed (Fig. 4b), indicating that client recruitment and complementation can rapidly occur in a minute time scale. When Homo-FC alone was expressed (without 0.5 N FUS) as a client, green signals inside condensates were hardly observed, likely due to limited recruitment. We next fused full-length 1.0 FUS to Homo-FC for client recruitment. 1.0 FUS–Homo-FC clearly showed stronger complementation signals than 0.5 N FUS–Homo-FC (Fig. 4b). As 1.0 FUS–1.0 FUS interactions should be stronger than 0.5 N FUS–1.0 FUS interactions, 1.0 FUS–Homo-FC can be recruited more effectively and quickly than 0.5 N FUS–Homo-FC. The data suggest that the recruitment strength of clients into condensates can be selectively controlled in cells by tuning scaffold–client interactions.

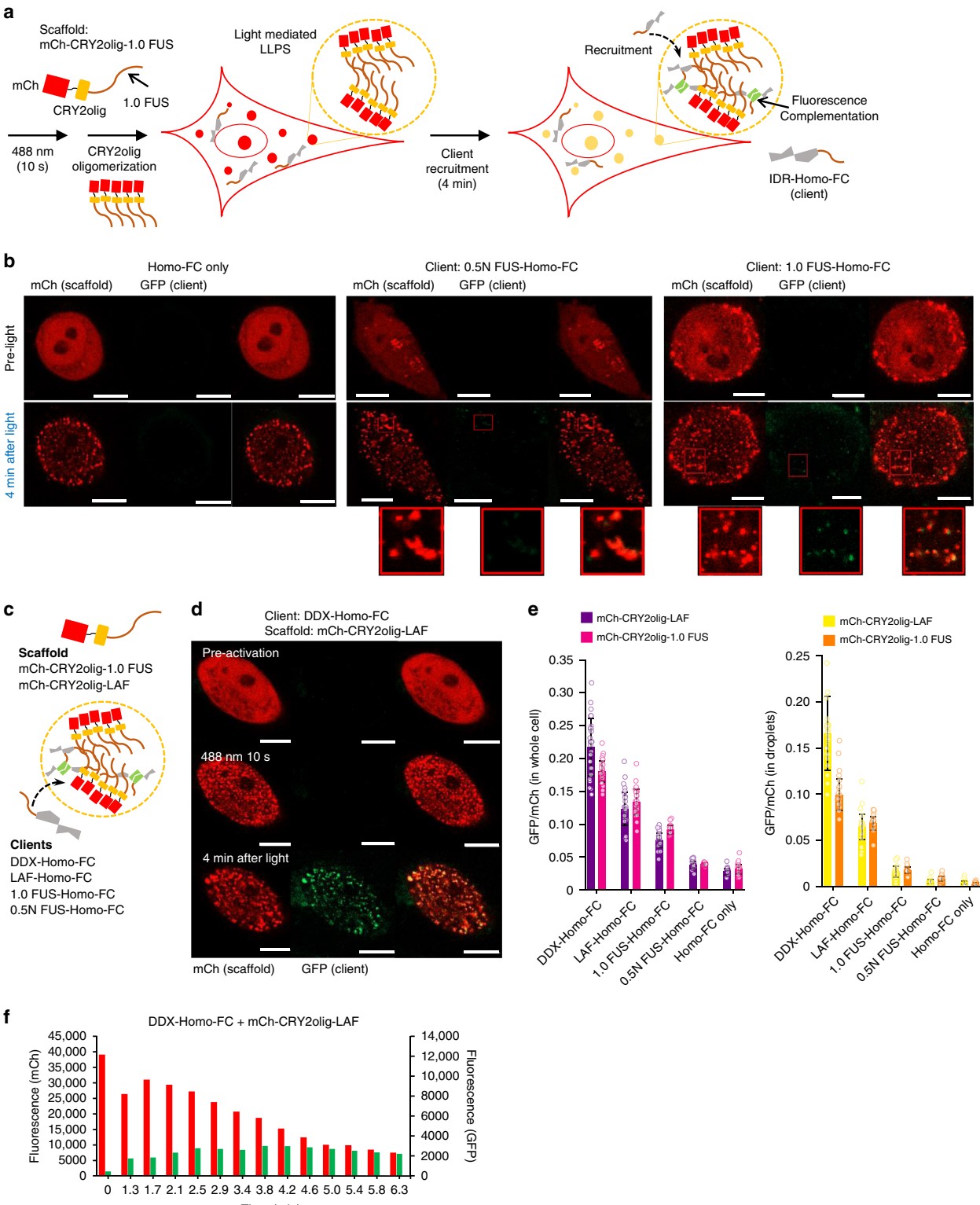

**Fig. 4 Real-time observation of fluorescence complementation inside light-induced IDR compartments. a** Schematic illustration of light-induced cellular condensate formation of mCh-CRY2olig-1.0 FUS, 0.5 N FUS–Homo-FC recruitment, and subsequent client complementation for (green) fluorescence turn-on. **b** Fluorescence images of cells expressing mCh-CRY2olig-1.0 FUS (scaffold) with client Homo-FC only (left), 0.5 N FUS–Homo-FC (middle), and 1.0 FUS–Homo-FC (right) before and after 10 s 488 nm light activation and 4 min incubation. **c** Schematic illustration of light-inducible cellular condensates with two kinds of scaffolds and four different Homo-FC clients with various IDPs. **d** Fluorescence images of cells expressing mCh-CRY2olig-LAF (scaffold) with client DDX-Homo-FC before and after 10 s 488 nm light activation, followed by 4 min incubation. **e** Total GFP(complemented Homo-FC)-to-mCh ratios for whole cells (left) or condensates (right) with various combinations of IDP scaffolds and clients. Data are presented as mean values with ±1 SD as error bars (n = 33 cells from three independent experiments). **f** Scaffold (mCh) and client complementation (GFP) signal changes inside condensates at various time points after 10 s light activation. Scale bars: 10 μm.

We next tested different combinations of IDPs as client-recruiting tags and condensate-forming scaffolds. Two other condensate-forming IDRs from LAF-1 (LAF) and Ddx4 helicase (DDX) proteins were employed[34,35]. We recently reported that clients with a LAF or DDX IDR tag were strongly recruited to diverse IDR condensates, whereas recruitment of clients with FUS was significantly less effective[24]. Four Homo-FC clients (with DDX, LAF, 1.0 FUS, or 0.5 N FUS) and two scaffold constructs (mCh-CRY2olig-1.0 FUS and -LAF) were prepared (Fig. 4c). Among eight possible client/scaffold combinations, DDX-Homo-FC with LAF condensates showed the strongest complementation signals (Fig. 4d, e and Supplementary Fig. 8). In particular, effective recruitment of DDX-Homo-FC and LAF-Homo-FC compared to 1.0 FUS–Homo-FC (particularly inside condensates) is consistent with the previously reported in vitro recruitment data[24]. The data clearly demonstrate that protein recruitment and complementation can dramatically vary by the nature of clients and scaffolds. Real-time live-cell images showed rapid increases of complementation signals within the first 4 min after condensate formation (Fig. 4f and Supplementary Fig. 9a). Given the required maturation time for FC[36], client recruitment of IDR condensates can be considered as a nearly instant process for strong clients. A portion of IDR condensates could also rapidly move inside cells even during a short time gap between sequential mCh and GFP imaging, which caused sporadic imperfect mCh-GFP co-localization (Supplementary Fig. 9b). FRAP analyses indicate that both mCh-CRY2olig-LAF and -1.0 FUS condensates have liquid-like properties, while their mobile fractions and diffusivities were lower than those of 2.0 FUS condensates (Supplementary Fig. 10).

Despite rapid and strong probe complementation inside CRY2olig-IDP condensates, client probe enrichment inside condensates was very low (PCs < 1.6) (Supplementary Fig. 11). For example, DDX-fused GFP (the strongest client model) signals inside CRY2olig-LAF condensates were only 1.6-fold higher than signals outside of condensates. However, complementation signals of DDX-Homo-FC with LAF condensates at the lowest protein levels were significantly higher than those without condensates even at the highest protein levels (Supplementary Fig. 12). Again, it is possible that probe complementation inside condensates is significantly more effective than mere expectation from relative concentration consideration. To check if the observed behaviors are specific to our choice of CRY2olig, we also tested the Vivid (VVD) protein, which dimerizes (rather than oligomerizes like CRY2) by a 488 nm light[37]. Cells with VVD-fused mCh-1.0 FUS showed significant condensate formation (comparable to mCh-2.0 FUS) by 20 min light exposure (Supplementary Fig. 13), indicating slower LLPS than by CRY2. However, when 0.5 N FUS–Homo-FC (weak client) was co-expressed with light-inducible mCh-VVD-1.0 FUS, strong FC signals were observed inside condensates during 40 min incubation after light activation (Supplementary Fig. 14), again even in cells with the lowest protein expression levels (Supplementary Fig. 15).

**Real-time observation of FP interactions inside light-induced IDR compartments**. Although we observed strong FC enhancement inside cellular condensates, complementation is a unique protein interaction that is nearly irreversible, which might exaggerate client proximity within condensates. Thus, we investigated light-switchable, tetrameric FP Dronpa 145 N as a different (and more conventional) interaction-responsive probe. A portion of expressed Dronpa is in a fluorescently active tetrameric form, and it is inactivated by a 488 nm light, which strongly favors its monomeric form (Fig. 5a)[38,39]. In general, fluorescent tetramers and inactive monomers are in equilibrium, which can be shifted

by lights (488 nm and 405 nm). To monitor equilibrium changes of Dronpa inside condensates, DDX-fused Dronpa (the strongest client) was co-expressed with mCh-CRY2olig-LAF. DDX-Dronpa displayed strong green signals, indicating the presence of fluorescent tetrameric Dronpa (Fig. 5b, pre-light). When the cells were exposed to a 488 nm laser for 10 s, LAF condensates were instantly produced, and green Dronpa signals were simultaneously turned off, indicating that DDX-Dronpa proteins were mostly switched to monomers. During the next 4 min, strong green signals appeared again, particularly inside condensates, implying that (turned off) monomeric DDX-Dronpa probes are recruited into condensates and the Dronpa equilibrium shifts toward fluorescent tetrameric states, as Dronpa favors tetrameric forms when its concentration is high[40]. However, the actual concentration enrichment of recruited client molecules inside condensates was only marginal as demonstrated with the GFP-fused DDX client (Supplementary Fig. 11). These data clearly indicate again that tetramerization is more effective inside condensates than outside even when the probe concentration inside is not significantly higher (or even lower; Supplementary Fig. 16) than outside. As discussed with enhanced FC, in addition to slight concentration increases of clients, other IDR compartment environments must also significantly contribute to enhanced Dronpa monomer interactions into tetramers. It is also noteworthy that when Dronpa was expressed without DDX fusion or DDX-Dronpa was expressed without condensates, green signal enhancement was not observed (Fig. 5c), implying that DDX-mediated Dronpa recruitment and subsequent tetramerization require LAF condensates.

We also tested various client/scaffold IDR pairs as in the previous section to examine the efficiency of Dronpa tetramerization inside compartments as a function of recruitment tendency. Similarly, the tetramerization efficiency was also greatly varied by client IDRs, where 0.5 N FUS-Dronpa showed the weakest signal enhancement and DDX-Dronpa showed the strongest (Fig. 5d and Supplementary Fig. 17). Diffusion of tetrameric Dronpa clients ($t_{1/2} = 18.3$ s) was significantly faster than that of CRY2olig-IDR scaffolds ($t_{1/2} = 156$ s) with a higher mobile fraction (clients 86% and scaffolds 46%) (Supplementary Fig. 10 vs. 18). Several studies reported that clients show more rapid diffusion than scaffolds[24,41,42]. Nevertheless, the recovery time of DDX-Dronpa tetramers around CRY2olig-LAF condensates ($t_{1/2} = 18.3$ s) is vastly longer than that of cytosolic proteins ($t_{1/2} < 1$ s)[43]. Live-cell images also showed instant recruitment and tetramerization of Dronpa upon condensate formation within 2–4 min (Supplementary Fig. 19). The Dronpa probe was also applied to FUS-repeat condensates as demonstrated with the Homo-FC probe. When Dronpa was directly fused to length-varied mCh-FUS, Dronpa with longer 2.0 FUS formed clear cellular condensates, whereas Dronpa with 0.5 N FUS did not show any noticeable puncta (Supplementary Fig. 20a). In addition, strong green signals appeared rapidly only from mCh-2.0 FUS-Dronpa (particularly in puncta) but not from mCh-0.5 N FUS-Dronpa, consistent with the Homo-FC experiments (Fig. 2). When 0.5 N FUS-fused Dronpa (client) was co-expressed with mCh-2.0 FUS (scaffold), again, strong green signals appeared rapidly, particularly inside mCh-2.0 FUS condensates (Supplementary Fig. 20b, c).

Moreover, to further validate observed proximity enhancement with proteins that are naturally found together in the same cellular condensates, we examined two additional IDR-containing proteins, TIA1 and TAF15, both of which have been found with FUS in cellular stress granules[44–46]. IDRs of TIA1 and TAF15 were fused to Dronpa as clients and applied to light-inducible FUS. Strong green signals (by Dronpa tetramerization) appeared rapidly during the 4 min incubation after light-induced FUS

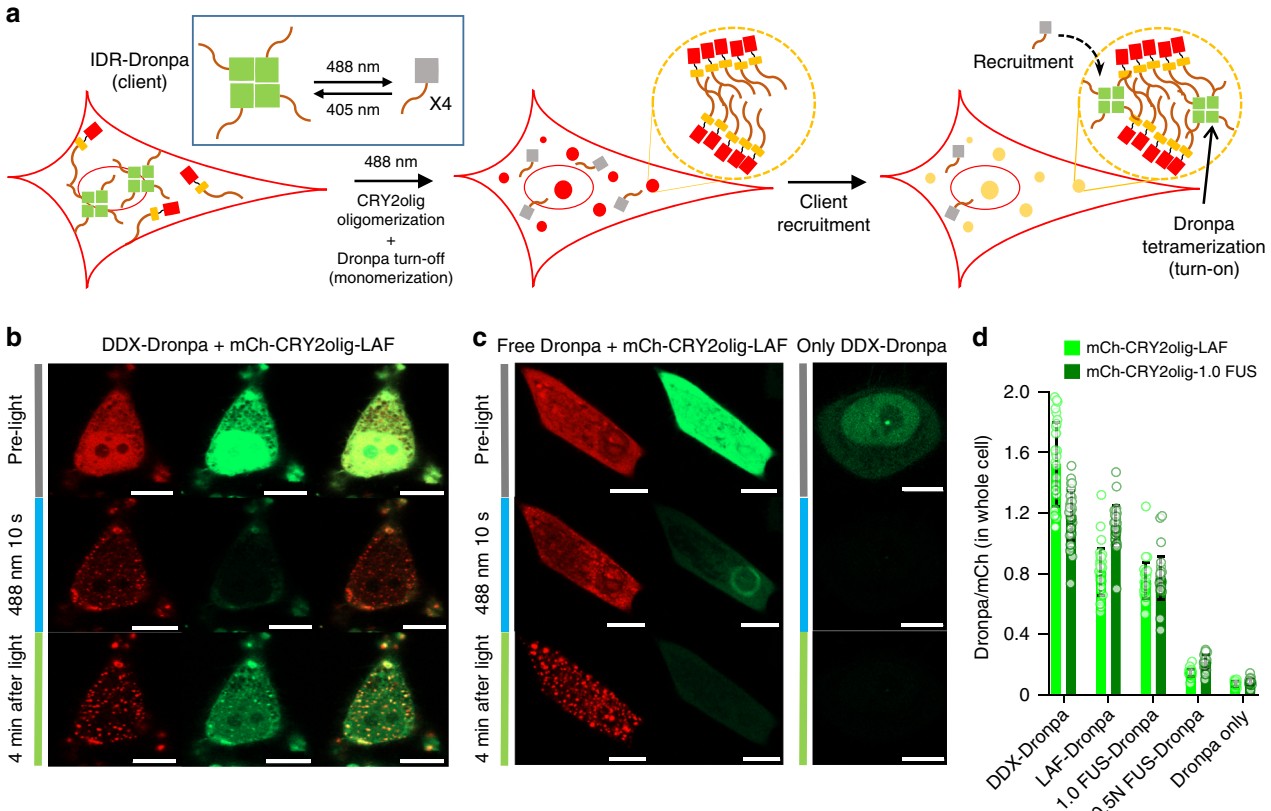

**Fig. 5 Real-time observation of FP interactions inside light-induced IDR compartments. a** Schematic illustration of light-induced condensate formation by mCh-CRY2olig-IDR oligomerization and simultaneous IDR-Dronpa client turn-off (monomerization), followed by IDR-Dronpa recruitment and subsequent Dronpa tetramerization for (green) fluorescence turn-on inside condensates. **b** Fluorescence images of cells expressing mCh-CRY2olig-LAF (scaffold) with client DDX-Dronpa before and after 10 s 488 nm light activation and 4 min incubation. **c** Fluorescence images of cells expressing mCh-CRY2olig-LAF (scaffold) with free Dronpa (left) or cells with only DDX-Dronpa without scaffold (right) before and after 10 s 488 nm light activation and 4 min incubation. Scale bars: 10 μm. **d** Total GFP (tetramerized Dronpa)-to-mCh ratios for whole cells with various combinations of IDP scaffolds and clients. Data are presented as mean values with ±1 SD as error bars (*n* = 33 cells from three independent experiments).

condensate formation (Supplementary Fig. 21), indicating that the proximity of the TIA1 or TAF15 client was also enhanced upon recruitment into FUS condensates. The signal enhancement ratios (Dronpa/mCh) were lower than those of FUS and DDX clients but higher than that of 0.5 N FUS (Fig. 5d vs. Supplementary Fig. 21b).

**Quantitative measurement of FP interaction enhancement inside IDR compartments**. Our intracellular condensate experiments under various conditions clearly indicate that client FP probes showed stronger interaction signals inside condensates than outside even when their concentrations are similar. To precisely determine the extent of client protein enrichment and additional interaction enhancement inside IDR droplets, we designed an in vitro IDR compartment model, where we can accurately tune and measure client concentrations. We produced droplets from LAF scaffolds by clustering biotinylated LAF with tetrameric streptavidin (STA), as previously reported[24]. The solution also contained monomerized (turned off) LAF-Dronpa client proteins, which would be recruited to LAF droplets, resulting in client enrichment and subsequent proximity-dependent Dronpa tetramerization (turn-on) (Fig. 6a). LAF-Homo-FC was prone to aggregation, and therefore, could not be used for this quantitative in vitro measurements. Client LAF-Dronpa was labeled with a Cy5 dye and, therefore, absolute LAF-Dronpa client concentrations (inside droplets) could be

determined from Cy5 signals (Supplementary Fig. 22a). Monomerized LAF-Dronpa could rapidly interact with each other to become fluorescently active tetramers in a concentration-dependent manner (Supplementary Fig. 22b). Therefore, we measured fluorescence signals of monomerized LAF-Dronpa solutions over a range of concentrations, ranging from 1 μM to 50 μM (Fig. 6b). We envisioned that Dronpa signals inside droplets can be compared with the expected Dronpa signals based on the actual global LAF-Dronpa concentration over the droplets (from Cy5 signals) to calculate the degree of proximity enhancement of LAF-Dronpa molecules.

When 1.0 μM (monomerized) client Cy5-LAF-Dronpa was mixed with droplets, the actual concentration of recruited Cy5-LAF-Dronpa inside droplets was determined to be 4.87 μM from the Cy5 signal, showing ~5-fold client enrichment (Fig. 6c). The expected Dronpa signal from 4.87 μM LAF-Dronpa is 818 (Fig. 6b). Surprisingly, however, the observed Dronpa signal from droplets was 15,582, which is 19 times higher than the expected signal (Fig. 6d). The data clearly indicate high proximity enhancement between recruited clients inside IDR droplets. The additional proximity enhancement effect was consistently observed with various client concentrations (Fig. 6d). We also examined the Dronpa client with truncated LAF (0.5 C LAF-Dronpa), which would have weaker interactions with LAF droplet scaffolds (thereby weaker recruitment). In fact, after LLPS with 2 μM client Cy5-0.5 C LAF-Dronpa, the actual Dronpa concentration inside droplets (Cy5) was only 4.18 μM (Fig. 6c).

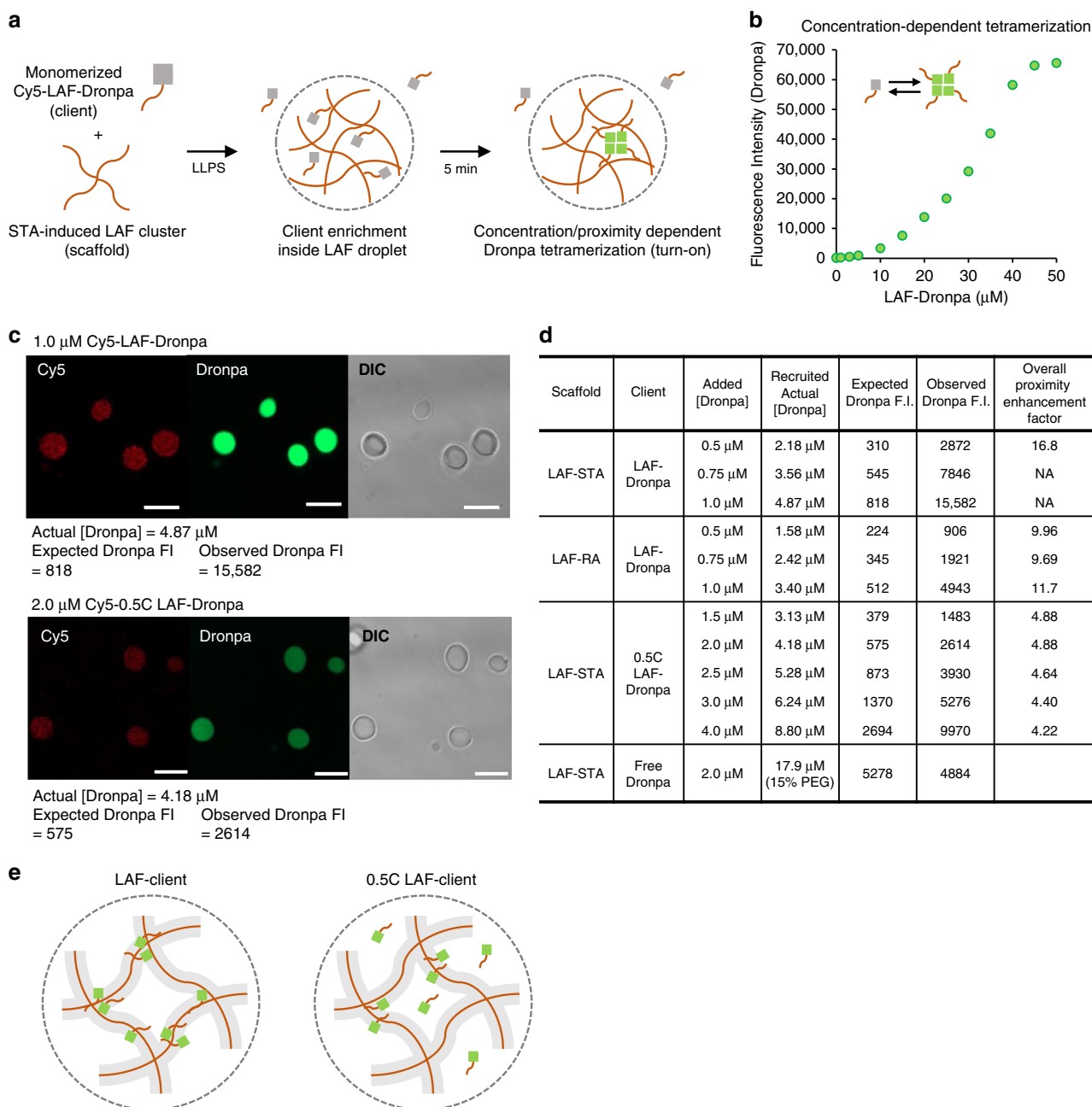

**Fig. 6 Quantitative measurement of protein interaction enhancement inside IDR compartments. a** Schematic illustration of LAF clustering-mediated droplet formation and monomerized Cy5-LAF-Dronpa client enrichment inside droplets, followed by proximity-dependent Dronpa tetramerization. **b** Dronpa fluorescence signals of concentration-varied LAF-Dronpa after 10 s 488 nm light activation for Dronpa monomerization and 4 min incubation for concentration-dependent tetramerization. **c** Cy5 and Dronpa fluorescence images of LAF droplets with mixed 1 μM client LAF-Dronpa (top) or 2 μM 0.5 C LAF-Dronpa (bottom). Calculated actual (Cy5) client concentrations and expected/observed Dronpa fluorescent intensities (FI) are indicated below the images. Scale bars: 10 μm. **d** Overall proximity enhancement factors with added Dronpa concentration, recruited actual Dronpa concentration inside droplets, expected Dronpa FI based on the actual [Dronpa], and observed Dronpa FI for various scaffold and client systems. **e** The proposed mechanism of proximity enhancement inside IDR droplets for LAF-Dronpa (left; strong scaffold–client interaction) or 0.5 C LAF-Dronpa (right; weak scaffold–client interaction). Potential client occupying volumes around scaffold protein networks are indicated in gray.

More importantly, at the similar actual Dronpa concentration inside droplets, LAF-Dronpa exhibited over three times higher tetramerization signals than 0.5 C LAF-Dronpa (Fig. 6d and Supplementary Fig. 22c). We also tested free Dronpa without IDR (LAF) fusion to examine the client proximity change in the absence of client interactions to condensate scaffold proteins. Interestingly, the proximity-dependent Dronpa signal inside condensates was nearly identical to the expected signal based

on the actual Dronpa concentration (Supplementary Fig. 23), indicating no client proximity enhancement inside condensates. It is clear that scaffold–client interactions strongly affect both client recruitment and additional proximity enhancement inside IDR droplets.

We also prepared LAF droplets by clustering LAF scaffolds with dimeric rhizavidin (RA) instead of tetrameric STA, with an aim to alter droplet properties, particularly a scaffold protein

density. We measured LAF (scaffold) concentrations (density) inside condensates. The LAF concentration of LAF-STA droplets was 252 μM, while the LAF concentration of LAF-RA droplets was 182 μM (Supplementary Fig. 24a), indicating that LAF is less dense in RA droplets. When Cy5-LAF-Dronpa was mixed with both droplets, the actual Dronpa concentration inside RA droplets was lower than that inside STA droplets (Fig. 6d and Supplementary Fig. 24b). The client-recruiting power of less-dense RA droplets (three- to fourfold enrichment) is clearly weaker than that of STA droplets (~5-fold enrichment). In addition, even when similar amounts of LAF-Dronpa was recruited into RA or STA droplets, proximity-dependent Dronpa signals were clearly higher (~1.5-fold) for LAF-Dronpa in STA droplets than in less-dense RA droplets (Fig. 6d).

One plausible working mechanism of proximity enhancement is that recruited (enriched) clients are further localized around IDR scaffold protein networks due to transient scaffold–client interactions, which can greatly increase the proximity of recruited client molecules (Fig. 6e). Hence, weaker client–scaffold interactions or less frequent interactions in less-dense droplets would yield less localization of clients around scaffolds, leading to weakening of proximity enhancement. We suggest that clients are heterogeneously distributed inside condensates, and it is more likely that they locate around scaffolds. The population of clients that are close to scaffolds can be increased by stronger client–scaffold interactions or by a higher scaffold density. Although it is impossible to experimentally determine the relative distribution of client molecules inside a droplet, we calculated the overall proximity enhancement factor from the actual global client concentration and the observed Dronpa signal (Fig. 6d). When clients are locally distributed around scaffolds inside condensates, we can simply assume that clients occupy only a fraction of the whole condensate volume ($f_{occupy}$) and feel concentrated in this smaller volume (proximity enhancement). Here, $f_{occupy} = V_{occupy}/V_{total}$, where $V_{occupy}$ is the client occupying volume and $V_{total}$ is the total droplet volume. The overall proximity enhancement factor is defined as $1/f_{occupy}$. It is noteworthy that although our two-state ("all-or-none") model is a simplification of the real concentration distribution inside condensates, the model can provide useful quantitative variables (see below) while it is highly challenging (if not impossible) to experimentally determine the real distributions.

The actual global client concentration inside a droplet is calculated as

$$c_{actual} = c_{occupy} f_{occupy} \qquad (1)$$

Also, the observed Dronpa signal of the same droplet is calculated as

$$I_{observed} = I(c_{occupy}) f_{occupy} \qquad (2)$$

Here, $c_{occupy}$ denotes the client concentration in $V_{occupy}$ (called the effective concentration) and $I(c)$ indicates the Dronpa signal intensity at client concentration $c$. From our experimental data ($c_{actual}$, $I_{observed}$, $I(c)$), these equations can be numerically solved to determine $f_{occupy}$ and its consequent overall proximity enhancement factor for each experimental system. Surprisingly, the proximity of LAF-Dronpa was increased over 16-fold in STA droplets. The enhancement factors could not be calculated at high LAF-Dronpa concentration, as $c_{occupy}$ exceeded the experimental Dronpa concentration limit (50 μM) with this high enhancement factor. Proximity enhancement factors of LAF-Dronpa in RA droplets and 0.5 LAF-Dronpa in STA droplets were ~10 and ~4.5, respectively, which strongly supports the idea that client–scaffold binding is a main determinant for client proximity enhancement. Calculated enhancement factors were largely consistent over

various client concentrations, again supporting the validity of our proposed working mechanism model.

The overall proximity enhancement factor can also be used to estimate the effective binding free energy between the client and the scaffold network. We adapt a simple assumption that the volume reduction ($f_{occupy}$), or the decrease in translational degrees of freedom, is exactly compensated by the effective binding free energy per each client molecule, denoted as $\varepsilon$:

$$\varepsilon - k_B T \ln V_{occupy} = -k_B T \ln V_{total} \qquad (3)$$

Hence, from the volume ratio $f_{occupy}$, we can estimate

$$\varepsilon = k_B T \ln f_{occupy} \qquad (4)$$

Binding free energies of Cy5-LAF-Dronpa and Cy5-0.5 C LAF-Dronpa are estimated as ~1.7 and ~0.9 kcal/mol, respectively. The estimated binding free energy is proportional to the length of IDR as expected, which implies that additive IDR–IDR interactions are the major factor of proximity enhancement.

Lastly, we examined the diffusivity change of condensate scaffolds by client recruitment. Interestingly, FRAP recovery profiles clearly indicated that a scaffold mobile fraction was reduced by adding client proteins (Supplementary Fig. 25). For example, when 2 μM LAF-Dronpa (client) was mixed with LAF (scaffold) condensates, the scaffold mobile fraction was reduced from nearly 90% (no client) to 50%. The mobile fraction of client proteins was also slightly reduced upon recruitment. The mobile fraction reduction increased as the added client concentration increased. In addition, the LAF-Dronpa client (stronger scaffold binder) was more effective for mobile fraction reduction than 0.5 C LAF-Dronpa (weaker scaffold binder) (Supplementary Fig. 25a). Dronpa-free LAF clients were also shown to reduce both scaffold and client mobile fractions, but in a slightly lesser degree (Supplementary Fig. 25d, e). It is not clear how small amounts of client proteins influence the scaffold protein diffusivity inside condensates, but we suggest that client molecules may "glue" different scaffold molecules by transient scaffold–client interactions. Further studies will be needed to precisely correlate the reduced protein mobility with the client proximity enhancement.

## Discussion

In this work, we were able to quantitatively monitor enhanced FP complementation and/or interactions inside various IDR condensates in cells. Tandemly repeated and light-clustered IDRs produced membrane-less cellular compartment models with diverse formation rates, scaffold densities, diffusivities, and client recruitment abilities. In particular, we investigated a wide range of client recruitment degrees by using IDR–IDR interactions between different combinations of client/scaffold IDRs. A large fraction of the human proteome comprises IDRs[47], and cellular condensates normally contain many different IDRs. IDR–IDR interactions are likely one of the key cellular strategies for effective compartment formation and selective client recruitment. Single-component clients also allowed simple two-construct transfection (a client + a scaffold) for reliable two-color imaging with less concern for relative expression variations. In fact, overall protein expression levels varied fairly widely, often over fivefold, and we were able to examine diverse client concentrations inside and outside of condensates in live cells. Still, there are several caveats in our method: FP complementation is nearly irreversible and, thereby, we might observe accumulated signals. In addition, although the Dronpa probe is likely reversible, the mechanism of tetramerization-dependent fluorescent signal generation is not fully understood yet, and a turn-off step by light is necessary. Lastly, although client concentrations and condensate enrichment degrees could be accurately measured in vitro, the cellular

environment is different from the test tube condition and we should be careful when comparing the in vitro and in cell data.

We found several interesting behaviors of IDR-containing clients inside membrane-less compartments. First and most importantly, interior environments of condensates greatly enhance FP complementation and interactions. To date, enhancement of client reactions inside compartments has been mostly explained by simple client concentration enrichment[10]. However, we discovered that the proximity between enriched clients is further enhanced, possibly by client localization around scaffold protein networks inside condensates via scaffold–client interactions. Slower or less free client movement due to frequent client–scaffold interactions might also contribute to enhanced protein interactions by lowering entropic penalties. It should be stressed that, due to the discrepancy between effective and actual global concentrations, the client PC should be carefully used when estimating the client concentration inside droplets. Although the PC can serve as a proxy for the actual concentration, the effective concentration (which determines scaffold–client and client–client interactions) can be much higher, as shown in this work.

Second, IDR-mediated client recruitment can be an instant process in a minute time scale. Upon condensate formation, client interaction signals reached their maximum within 4 min. As this fast recruitment was reproduced in vitro, simple diffusion along the chemical potential gradient (rather than active processes) can be considered as the dominant mechanism for recruitment. Within 1 min, a protein with a diffusion coefficient of $10^{-6}$ cm$^2$/s[48] can diffuse up to sub-mm even with no external chemical gradient: $\langle x \rangle \sim \sqrt{6Dt} \sim 190$ μm for $t = 60$ s. Lastly, we observed a huge change in client recruitment behaviors and client–client interactions inside condensates depending on the nature of the given IDR pairs. Cells can selectively enrich diverse clients inside compartments with varying degrees of recruitment by choosing specific sets of IDRs among a huge pool of the IDR proteome.

Although we have developed several IDR condensate-client systems to examine quantitative aspects of client recruitment and interaction enhancement, more diverse model systems are required to further elucidate distinct behaviors of diverse biomolecular processes inside membrane-less organelles. For example, building liquid compartments with precisely controllable properties (e.g., density and diffusivity) will be valuable to understand the roles of specific condensate environments for client interactions. It would be useful to develop methods for quantitative monitoring (including sensor probes) of various other biomolecular processes, such as enzyme reactions, inside condensates. Strategies to selectively recruit multiple clients and vary (and accurately measure) client/scaffold concentrations would also be helpful.

## Methods

**Gene construction.** Genes for FPs (mCh, Dronpa 145 N, charged superfolder GFP), IDR proteins (DDX4, LAF, FUS, TIA1, and TAF15) and the light-activatable proteins (Dronpa 145 N, CRY2olig, CRY2, VVD) were synthesized by Bioneer (Daejeon, South Korea). All protein constructs for cell studies were cloned into the pcDNA3.1 (+) vector (Invitrogen). Biotinylated IDR protein genes were cloned into the pProExHTα expression vector (Invitrogen) and other recombinant protein genes were cloned into the pET-21a expression vector (Invitrogen). All protein sequences and primer sequences for gene cloning are listed in Supplementary Information.

**In vitro protein preparation.** Recombinant STA, RA, biotinylated IDR proteins, and LAF-fused Dronpa proteins were all prepared as previously described[24]. Briefly, biotinylated IDR proteins were expressed in AVB101 (Avidity) cells, and other proteins were expressed in *Escherichia coli* BL21 (DE3). All proteins were purified by Ni-IDA columns (BioProgen, Daejeon, South Korea). Avidin proteins were produced as inclusion bodies, which were dissolved in 6 M Guanidine hydrochloride (GuHCl) and 50 mM Tris-HCl pH 8.0 for overnight at 4 °C. For refolding, denatured STA and RA proteins were diluted dropwise into PBS and filtered through a 0.22 μm membrane filter (Stericup® Quick Release, Millipore

Express® PLUS), followed by overnight incubation at 4 °C before column purification. IDR-fused proteins were stored in a buffer containing 200 mM NaCl, 50 mM Tris pH 8.0, and 10% Glycerol. 1.0 LAF-Dronpa and 0.5 C LAF-Dronpa were labeled with NHS-Cy5 (Lumiprobe) by mixing proteins with a dye in a 1 : 0.5 protein/dye ratio. The mixed solutions were incubated for 40 min at 25 °C with shaking, and Cy5-labeled LAF-Dronpa was purified by a PD10 desalting column (Sephadex™ G-25 M, GE Healthcare). Protein concentrations were determined by Bradford assays and OD$_{280 nm}$ measurements with the Beer–Lambert equation.

**In cell studies.** A549, HEK293T, and HeLa cells with fewer than 20 passages were maintained in Dulbecco's modified Eagles' medium (Gibco, USA) supplemented with 10% fetal bovine serum (Gibco, USA) and 1% penicillin–streptomycin (Gibco, USA) at 37 °C under 5% CO$_2$ in humidified atmosphere. For light-induced phase separation, transfected cells with VVD containing constructs were illuminated to 488 nm light by using a mounted UV LED (470 nm/25 nm (Band width), 2.2 mW cm$^{-2}$, Thorlab) for 20 min, followed by incubation at 37 °C under 5% CO$_2$. Cells for control experiments were maintained in dark with aluminum foil wrapped. For real-time observation with CRY2 constructs, transfected cells were illuminated to 488 nm light by using a 488 nm laser (0.07 μW) of a confocal microscope (Carl Zeiss, LSM 800) for 10 s. Cell viability was examined by a tetrazolium-based calorimetric assay (MTT 3-(4,5-dimethylthiazol-2-yl)-2,5-diphenyltetrazolium bromide) assay). HeLa cells with fewer than 20 passages were seeded onto 96-well plates at a density of $1 \times 10^4$ cells per well and incubated for 18 h. Cells were transfected and incubated in the medium for 20 h before a MTT analysis.

**Fluorescence recovery after photobleaching.** Laser power was set to be 70% for both 488 nm (GFP) and 561 nm (mCh) lasers during 10 s bleaching. Fluorescence intensities during the pre-bleach, bleach, and post-bleach sequences were measured in FIJI (Fiji Is Just ImageJ) to generate recovery curves. The overall mCh signals in a cell (or the nucleus) was reduced by photobleaching. To compensate this unintended bleaching, we measured mean fluorescent intensity ratios between the region of bleaching (ROB) and the total nucleus ($I_{ROB}/I_{TOT}$). We normalized that $I_{ROB}/I_{TOT}$ equals 1 before bleach and 0 after bleach to minimize biases from whole condensate bleaching. Then the data were fit to the first-order exponential $f(t) = A(1 - e^{(-t/\tau)})$ for calculating half-life and mobile fraction; $f(t)$ is the normalized fluorescence at time $t$ after bleach ($f(t) = 1$ pre-bleach, $f(t) = 0$ at $t = 0$), A is the amplitude of recovery. Half-lives ($t_{1/2}$) of fluorescence recovery was calculated by using $t_{1/2} = (\ln 2) \times \tau$.

**Image analysis and data collection.** Fluorescence microscopy studies were performed with a confocal microscope (Carl Zeiss, LSM 800) using a ×100 oil objective lens. The cellular condensate formation ratio was calculated by dividing the sum of the mCh fluorescence intensities of generated droplets (>0.01 μm$^2$) by the total mCh intensity of a whole cell. The GFP/mCh or Dronpa/mCh ratio was calculated by dividing the total intensity of GFP or Dronpa of a whole cell or a droplet (>0.01 μm$^2$) by the total mCh intensity. Cells for data analysis were randomly selected ($n = 30 \sim 33$ cells, three independent experiments) with varied protein expression levels. Zen 2.5 (Carl Zeiss) and ImageJ (National Institutes of Health) were used for measuring fluorescence intensities and sizes of droplets. Microsoft Excel 2016 and Origin 2018 (Originlab) were used for graphing and statistical analyses.

**In vitro analysis of Dronpa interactions inside the droplets.** Fluorescence intensities (Cy5 and Dronpa) of Cy5-LAF-Dronpa (or Cy5-0.5 C LAF-Dronpa) with concentrations from 1 to 50 μM were measured with the identical confocal imaging condition (Dronpa 488 nm: 11.13% and Cy5 640 nm: 4.5% with the fixed z-axis) on a hydrophobic glass surface[24]. For hydrophobic coating of glass surfaces, slide glasses (Marienfeld) and cover glasses (Duran) were immersed to the piranha acid solution (95% Sulfuric acid 450 mL + 35% Hydrogen peroxide 150 mL) and incubated for 1 h at 60 °C. The glasses were serially washed by distilled water and acetone, and blown by nitrogen gas. The glasses were immersed to the coating solution (5 mL 3-(Trimethoxysilyl)propylmethacrylate, 10 mL acetic acid, and 35 mL acetone) and incubated with shaking on an orbital shaker (SH30, FINEPCR) at 25 °C for 2 h. The glasses were again washed by acetone and distilled water, and blown by nitrogen gas. The glasses were stored in a dry keeper (SANPLATEC, Co.) and used in 48 h.

Monomerized (turned off) Dronpa proteins were prepared by illuminating protein solutions with a 488 nm laser (100% power) for 20 min until fluorescent intensities became less than 200 at 488 nm (11.3%) excitation. To obtain the concentration-dependent tetramerization curve of LAF-Dronpa, Dronpa signals were measured after 4 min incubation of monomerized LAF-Dronpa. For in vitro proximity enhancement assays, STA (or RA) (60 μM) was mixed in a 1 : 1 molecule ratio with biotinylated LAF proteins, and the indicated concentration of monomerized LAF-Dronpa client protein was added. LLPS was induced by adding 1.5% weight PEG (molecular weight 8000 Da, lipopolysaccharide solution) (15% PEG for free Dronpa experiments). Mixed solutions were maintained for 5 min in tubes. For confocal analysis, 4 μL of solutions with droplets were dropped on a hydrophobic slide glass and covered by a cover glass for confocal analysis. Approximately 33 droplets were randomly selected for concentration determination. To determine scaffold LAF concentrations of STA- and RA

droplets, small ubiquitin-like modifier (SUMO)-fused LAF without biotinylation was labeled with Cy3 with a protein:dye ratio = 1 : 0.5 (reaction condition). Cy3 signals of Cy3-SUMO-LAF with concentrations from 10 to 160 μM were measured, and LAF concentrations inside droplets were calculated by extrapolation of this Cy3 curve as a function of LAF concentration.

**Statistics and reproducibility**. All number or replicates ($n$) and statistical analyses are indicated in corresponding figure legends. All cellular and in vitro assays were conducted at least three times, and similar results were seen in all independently performed experiments.

**Reporting summary**. Further information on research design is available in the Nature Research Reporting Summary linked to this article.

## Data availability

Data supporting the findings of this manuscript are available from the corresponding author upon reasonable request. A reporting summary for this Article is available as a Supplementary Information file. Source data are provided with this paper.

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

## Acknowledgements

This work is supported by Samsung Science & Technology Foundation (SSTF-BA1501-09) and the National Research Foundation of Korea (NRF) grant funded by MSIT (Y. Jung: NRF-2019R1A2C2008558, J.-M.C.: 2020R1I1A1A01070805). D.S. is funded by

the Basic Science Research Program (2019R1A6A1A10073887) from the Ministry of Education of the Republic of Korea through the National Research Foundation of Korea.

## Author contributions

Y. Jung, D.S., and J.-M.C. designed the study. D.S. conducted the in cell experiments. Y. Jo and D.S. conducted the in vitro experiments. J.-M.C. conducted the theoretical analysis. Y. Jung, D.S., and J.-M.C. drafted and edited the manuscript.

## Competing interests

The authors declare no competing interests.
