## [Peer Review File · Nature Communications]

REVIEWER COMMENTS

Reviewer #1 (Remarks to the Author):

Biomolecular condensates are cellular compartments that concentrate proteins and nucleic acids without surrounding membranes. Although condensates are associated with many aspects of cell biology, mechanisms of their formation and function are not well understood. A key parameter in assessing condensate function is their composition, which is often measured by partition coefficient (a ratio of protein concentration within vs. outside of condensates). The hypothesis is that molecules that are concentrated together will interact more frequently, which will exert functional benefit. In this manuscript, Jung and colleagues examine the first part of this idea using a combination of oligomerizing protein systems and fluorescence-based proximity readouts. They report that the interaction between molecules appear significantly more enhanced within condensates than what is expected from their partition coefficient. This is a valuable insight and therefore would be of interest to the readership of Nature Communications. The manuscript is written in a way that is easy to follow and the tools that the authors use to address this question is viewed as a strength. However, several limitations in the approach and presentation weaken their conclusion and I recommend that outlined issues must be addressed before publication in Nature Communications.

Major comments:

- The goal of the study is to look at the relationship between clients (molecules that partition into the condensates) and scaffolds (molecules that 'build' the condensates). The authors examine this using model protein systems -- intrinsically disordered regions of a few known phase-separating proteins, DDX FUS and LAF. These proteins do not normally interact in cellular condensates to my knowledge. Moreover, the first half of the study is done using different versions of a single protein, FUS. Accordingly, the results can only be suggestive of what may occur in cellular condensates. The conclusions will be significantly strengthened by validating the findings with proteins that are found together in cellular condensates (if possible using known scaffolds and clients). This may also provide insight into whether enhanced client proximity within condensates have important consequences for their function. Otherwise, the conclusions and discussions should be adjusted to reflect these limitations.
- Similarly, the conclusion that scaffolds limit client movement and thereby enhance client-client proximity within condensates is drawn from theoretical prediction. Testing this experimentally would be important, as this would demonstrate whether enhanced client proximity has a functional consequence. For example, super-resolution microscopy or EM may be able to show this, particularly in vitro.
- The main tools used in the study are two types of fluorescent proximity 'sensors'. The advantage of using these tools (over others like NMR or FCS) would be the ability to visualize client interactions in cells. However, the tools have important limitations. One of the sensors (FC) dissociates slowly and thus likely over-exaggerates client proximity within condensates. The dissociation of the second tool, Dronpa, is more controllable and should be a more accurate measure. Unfortunately, this tool was only used in the last figure. Moreover, it would be important to further validate their findings using another method (e.g. FCS) or adjust conclusions to reflect these limitations.
- The study mentions one possible functional consequence of enhanced client proximity: increased reaction kinetics. However, the exclusion of non-client molecules may also work similarly. For instance, 'repelling' forces between clients and excluded components may be enhanced if the client 'effective concentration' is higher in condensates. Another consequence may be in tuning the biophysical properties of the condensates. Testing these ideas would further strengthen the manuscript.
- Many of the conclusion would require measuring the ratio between client concentration and proximity read-out (as done in Figure 5), rather than the ratio between scaffold and client.

Minor comments:

- Title: Two aspects of client proximity enhancement inside cellular membrane-less compartments –

the two aspects would not be immediately clear to readers. Please consider using a more descriptive title.

- Manuscript organization: Many important controls and data were tucked away in supplement, making the manuscript difficult to follow and evaluate. For example, please consider moving results such as Figure S1, S3a, S4d, S9b into the main figure.
- Figures: Cell images are often too small to be able to evaluate. Please zoom in on all images and include insets focusing on a few condensates. In addition, please make scale bars clearly visible and consider using greyscale to enhance contrast. In many of the images, the cells often look very rounded, which is unusual for HeLa cells. Please include a measure of cell viability during these experiments.
- Please include n of cells analyzed for all experiments in legend.
- Many images seem over-exposed (particularly mCh) and GFP appears to not perfectly colocalize with the mCh signal as a result. Please show a line-profile or similar to show whether this is true.
- The term 'effective concentration' seems confusing and potentially misleading, given that the tools measure the proximity of molecules to each other. Please define this term explicitly or replace it with proximity.
- Please state the sensitivity/proximity range of the FC and Dronpa. This information is crucial to assess how direct the interactions reported by the tools are likely to be.
- Please avoid vague statements that appear throughout the manuscript. E.g. "how specific clients are selectively enriched and distinctly react"; "biomolecular behaviors inside these liquid compartments have been largely unexplored."; "understanding of biomolecular behaviors inside membrane-less compartments"; Many but not all "Membrane-less compartments show remarkable liquid-like properties"
- Correct "multivalent interactions between repeated proteins" on page 2 to "repeated folded protein domains".
- To support the statements on page 2, the following reference seems very relevant. Kim, Tsang...Forman-Kay 2019 demonstrated translational control by differential partitioning of components.
- Based on images in 2C, it seems that a significant portion of the nucleus is bleached for FRAP measurements, not the individual condensates. This may be why the recovery times are significantly slower compared to what has been reported for FUS.
- Please include DAPI to orient us in 2E.

Reviewer #2 (Remarks to the Author):

This is an interesting paper with an extensive amount of work for making condensates with a variety of different molecular tools, and correspondingly measuring or inferring the degree of concentration of clients in the scaffolds. I will not summarize all the molecules and clients that were used here, although I would say the presentation of the paper would be greatly included if the authors themselves provided a table of LLPS forming molecules, clients, and a summary of the keep result with this combination.

The point of the paper that the authors most want to make is that the effective interaction between clients is enhanced by their proximity in the droplets beyond their simple concentration. This would imply that trapping the clients in a local volume would enhance their interaction. A simple energy model to this effect is provided, and suggested that for molecules with a binding one can deduce the effective constrained volume.

There are two systems used to describe this effective increase in client interactions. One, which is less convincing, involves homoFC constructs. The other, which is more convincing, involves the recruitment of LAF-Dronpa-Cy5 clients to LAF droplets, The cy5 measures the actual concentration

while the Dronpa tetramerization measures the enhancement effect, Although I question whether the enhancement is due to the tetrameric interactions (versus the the dimerization required for homoFc), there is a clear indication of enhancement, as represented in Figure 5.

Other experiments presented here do not address the enhancement, but do add to the impressive ability of this group to engineer phase separation across a wide range of current protein chemistries. Overall, this paper is an impressive tour de force and I favor publication.

Reviewer #3 (Remarks to the Author):

This is an interesting and very important study that adds significantly to the field and will have a noticeable impact. The manuscript is well-written and concise. It reports impressive results of a set of well-thought and masterly performed experiments that shed light on the molecular mechanisms of biological liquid-liquid phase transitions (LLPTs) and formation of membrane-less organelle (MLOs). The authors showed that the operational effective concentrations inside the MLOs (i.e. concentrations defining interactions between the clients and based on the measured client-client interactions in MLOs) could significantly exceed the actual concentrations of the compounds of MLOs. These effect was defined as the client proximity enhancement by the MLOs. The authors also proposed that such client proximity enhancement can be defined by a least two mechanisms, such as selective recruitment of the specific clients into MLOs and specific co-localization of clients (or MLO passenger proteins) in close proximity to the MLO-forming scaffold proteins (or LLPT drivers), leading to further increase in local client concentrations. Obviously, these observations re crucial for better understanding of basic principles driving biological LLPTs and related to the MLO biogenesis. However, there are two issues that need to be addressed.

1) The authors briefly mentioned in introduction section that recent studies indicated that "multivalent interactions between repeated proteins or intrinsically disordered proteins/regions (IDPs/IDRs) [4] can drive liquid-liquid phase separation (LLPS), and that this is the major formation principle of compartmentalized biomolecular condensates (also termed droplets) [5]." The provided references to two publications here, 4: Dyson, H.J. & Wright, P.E. Intrinsically unstructured proteins and their functions. *Nat. Rev. Mol. Cell Biol.* 6, 197-208 (2005); and 5: Alberti, S., Gladfelter, A. & Mittag, T. Considerations and Challenges in Studying Liquid-Liquid Phase Separation and Biomolecular Condensates. *Cell* 176, 419-434 (2019). Although reference [4] represents one of the important reviews on IDPs, it does not discuss the roles of multivalent interactions between IDPs/IDRs in LLPS. However, there are several focused studies, where the role of such interactions between IDPs/IDRs are discussed. Furthermore, there are also several focused studies on the overall abundance of IDPs/IDRs in MLOs. In my view, these studies should be cited and discussed in this manuscript.

2) Although the authors mentioned that there is a selective recruitment of the specific clients/passengers into MLOs, they did not discuss mechanisms of such specific recruitment. However, there are several recent studies indicating that such recruitment can be a reflection of specific partitioning of clients/passengers into MLOs; i.e., a process similar to the partitioning of various substances into different phases of aqueous two-phase systems (ATPSs). In my view, these studies should be cited and discussed in this manuscript.

The changes in the revised manuscript were highlighted by giving the text a yellow background.

Reviewer 1

General remarks

Biomolecular condensates are cellular compartments that concentrate proteins and nucleic acids without surrounding membranes. Although condensates are associated with many aspects of cell biology, mechanisms of their formation and function are not well understood. A key parameter in assessing condensate function is their composition, which is often measured by partition coefficient (a ratio of protein concentration within vs. outside of condensates). The hypothesis is that molecules that are concentrated together will interact more frequently, which will exert functional benefit. In this manuscript, Jung and colleagues examine the first part of this idea using a combination of oligomerizing protein systems and fluorescence-based proximity readouts. They report that the interaction between molecules appear significantly more enhanced within condensates than what is expected from their partition coefficient. This is a valuable insight and therefore would be of interest to the readership of Nature Communications. The manuscript is written in a way that is easy to follow and the tools that the authors use to address this question is viewed as a strength. However, several limitations in the approach and presentation weaken their conclusion and I recommend that outlined issues must be addressed before publication in Nature Communications.

Major comments

- The goal of the study is to look at the relationship between clients (molecules that partition into the condensates) and scaffolds (molecules that ‘build’ the condensates). The authors examine this using model protein systems -- intrinsically disordered regions of a few known phase-separating proteins, DDX FUS and LAF. These proteins do not normally interact in cellular condensates to my knowledge. Moreover, the first half of the study is done using different versions of a single protein, FUS. Accordingly, the results can only be suggestive of what may occur in cellular condensates. The conclusions will be significantly strengthened by validating the findings with proteins that are found together in cellular condensates (if possible using known scaffolds and clients). This may also provide insight into whether enhanced client proximity within condensates have important consequences for their function. Otherwise, the conclusions and discussions should be adjusted to reflect these limitations.

Response: As the reviewer suggested, we examined two additional IDR-containing proteins, TIA1 and TAF15, both of which have been found with FUS in cellular stress granules (JCB 2016, 215, 313; Cell 2016, 164, 487; PLoS ONE 2012, 7, e46251). In fact, FUS, TIA1 and TAF15 are all well-known stress granule markers. TIA1 and TAF15 were fused to Dronpa as clients and applied to light-inducible FUS condensates as designed in Fig. 5. The Dronpa sensor system was applied since it is more controllable and accurate than Homo-FC as the reviewer noted in a later comment. After 10 sec light illumination (488 nm) to mCh-CRY2olig-FUS and TIA-Dronpa (or TAF-Dronpa) expressing cells, FUS condensates were instantly produced, and green Dronpa signals were simultaneously turned off (revised Fig. S21). Strong green signals appeared rapidly during the next 4 min, indicating that the proximity of the TIA or TAF client was also enhanced upon recruitment into FUS condensates. Normalized Dronpa signals (Dronpa/mCh) were lower than those of FUS and DDX clients but higher

than that of 0.5 FUS (revised Figs 5d vs S21b).

- In Results (Page 9, third paragraph): A paragraph was added to explain TIA and TAF-mediated client proximity enhancement with a figure and related references as discussed above.

- In Supplementary Information: A figure (Fig. S21), which shows TIA- or TAF-fused Dronpa interactions inside light-induced FUS compartments, was added.

- Similarly, the conclusion that scaffolds limit client movement and thereby enhance client-client proximity within condensates is drawn from theoretical prediction. Testing this experimentally would be important, as this would demonstrate whether enhanced client proximity has a functional consequence. For example, super-resolution microscopy or EM may be able to show this, particularly *in vitro*.

Response: We agree that our study lacked direct physical evidences for client-client proximity enhancement inside condensates (other than probe protein signal enhancement). However, measuring accurate physical distance distributions between client proteins inside and outside condensates is extremely challenging and has not been reported yet. As the reviewer suggested, super-resolution microscopy or EM might be able to offer this information, although many techniques must be properly developed. For example, protocols to measure physical distance distributions between proteins in solutions are not defined, and the labeling method for EM analysis of IDPs must also be developed. While we are unable to establish these new methods yet, we will continuously pursue for these new analysis tools. We discussed these much needed future works in Conclusion of the revised manuscript.

Still, to gain more insights and propose a more experimentally validated working mechanism on client proximity enhancement, we conducted additional quantitative *in vitro* assays under more diverse condensate conditions.

First, we tested free Dronpa without IDR (LAF) fusion to examine the proximity change of the Dronpa sensor protein inside condensates in the absence of client interactions to condensate scaffold proteins. Free Dronpa without LAF was slightly excluded from LAF condensates, and therefore, a high concentration of crowding reagent PEG (15%) was used to force Dronpa recruitment into condensates. Interestingly, the proximity-dependent Dronpa signal inside condensates was nearly identical to the expected signal based on the actual Dronpa concentration (revised Fig. S23), indicating no client proximity enhancement inside condensates. This data suggests that client-scaffold interactions are mainly responsible for the enhanced client proximity.

We next prepared LAF droplets by clustering LAF with dimeric rhizavidin (RA) rather than tetrameric streptavidin (STA), with an aim to alter a scaffold protein density. We measured LAF (scaffold) concentrations (density) inside condensates. [LAF] of LAF-STA droplets was 252 μM , and [LAF] of LAF-RA droplets was 182 μM (revised Fig. S24a), indicating that LAF is less dense in RA droplets. When Cy5-LAF-Dronpa was mixed with both droplets, [LAF-Dronpa] inside RA droplets was lower than that inside STA droplets (revised Figs 6d and S24). The client recruiting power of less-dense RA droplets (3-4 fold enrichment) is clearly weaker than that of STA droplets (~5 fold enrichment). More importantly, even when similar amounts of LAF-Dronpa was recruited into RA or STA droplets, proximity-dependent Dronpa signals were clearly higher (~1.5 fold) for LAF-Dronpa in STA droplets than in less-dense RA droplets (Fig. 6d).

Newly obtained free Dronpa and RA-droplet data as well as LAF-Dronpa and 0.5 LAF-Dronpa data consistently suggest that client localization around scaffold proteins by client-scaffold interactions is a major driving factor for client proximity enhancement inside condensates. In the absence of client-scaffold interactions (therefore, no client localization around scaffolds), proximity enhancement was not observed (free Dronpa). Stronger client-scaffold interactions (therefore stronger client localization around scaffolds) provided higher proximity enhancement (LAF-Dronpa vs 0.5C LAF-Dronpa). A higher scaffold density (therefore more frequent client localization around scaffolds) also provided higher proximity enhancement (STA droplets vs RA droplets).

To explain these observations, we suggest that clients are heterogeneously distributed inside condensates around scaffolds. The population of clients that are close to scaffolds can be increased by stronger client-scaffold interactions (or higher scaffold density), which yields client proximity enhancement. Although it is impossible to experimentally determine the relative distribution of client molecules inside a droplet, we were able to calculate the *overall proximity enhancement factor* from the actual global client concentration and the observed Dronpa signal. When clients are locally distributed around scaffolds inside condensates, we can assume that averagely, clients occupy only a fraction of condensates (f_{occupy}) and feel concentrated (proximity enhancement) in this smaller volume. Here, $f_{\text{occupy}} = V_{\text{occupy}}/V_{\text{total}}$, where V_{occupy} is the client occupying volume, and V_{total} is the total droplet volume. And we define the overall proximity enhancement factor as $1/f_{\text{occupy}}$.

Then, the actual global client concentration inside a droplet is calculated as

$$c_{\text{actual}} = c_{\text{occupy}} f_{\text{occupy}}$$

And the observed Dronpa signal of the same droplet is

$$I_{\text{observed}} = I(c_{\text{occupy}}) f_{\text{occupy}}$$

Here, c_{occupy} denotes the client (feeling) concentration in V_{occupy} , and $I(c)$ indicates the Dronpa signal intensity at client concentration c . Based on our experimental data (c_{actual} , I_{observed} , $I(c)$), this equation can be numerically solved to determine f_{occupy} . And as the reviewer suggested in a blow comment, instead of using the rather confusing term ‘effective concentration’, we provided a proximity enhancement factor ($1/f_{\text{occupy}}$).

The initially added client concentration, the actual client concentration recruited in a droplet, the expected Dronpa signal based on the actual client concentration, the observed Dronpa signal, and the calculated overall proximity enhancement factor were all listed in revised Fig. 6d. Surprisingly, the proximity of LAF-Dronpa was increased over 16 fold in STA droplets. The enhancement factors could not be calculated at high [LAF-Dronpa] since c_{occupy} exceeded the experimental Dronpa concentration limit (50 μM) with this extremely high enhancement factor. Proximity enhancement factors of LAF-Dronpa in RA droplets and 0.5 LAF-Dronpa in STA droplets were ~ 10 and ~ 4.5 , respectively, which strongly supports the idea that client-scaffold binding is a main determinant for client proximity enhancement. Calculated enhancement factors were largely consistent at various client concentrations, also supporting the validity of our proposed working mechanism model.

We thoroughly revised the last section (revised Fig. 6) with newly added data and calculated enhancement factors.

- In Results (Page 10, second paragraph): Multiple sentences “*We also tested free Dronpa without IDR (LAF) fusion to examine the client proximity change in the absence of client interactions to condensate*

scaffold proteins. Interestingly, the proximity-dependent Dronpa signal inside condensates was nearly identical to the expected signal based on the actual Dronpa concentration (Fig. S23), indicating no client proximity enhancement inside condensates.” were added to discuss proximity change of free Dronpa with an added data (Fig. S23).

- In Results (Page 11, second paragraph): A paragraph was added to discuss dimeric RA-induced droplets with an added data (Fig. S24).

- In Results (Page 11-12): Two paragraphs were added to introduce a newly proposed working mechanism of proximity enhancement as discussed above (Fig. 6d).

- In Results (Fig. 6d): A table (Fig. 6d), which contains overall proximity enhancement factors with added Dronpa concentration, recruited actual Dronpa concentration inside droplets, expected Dronpa FI based on the actual [Dronpa], and observed Dronpa FI for various scaffold and client systems, was added.

- In Supplementary Information: A figure (Fig. S23), which shows quantitative measurement of free Dronpa concentration and tetramerization inside LAF compartments, was added.

- In Supplementary Information: A figure (Fig. S24), which contains dimeric RA-mediated droplet related data, was added.

- The main tools used in the study are two types of fluorescent proximity ‘sensors’. The advantage of using these tools (over others like NMR or FCS) would be the ability to visualize client interactions in cells. However, the tools have important limitations. One of the sensors (FC) dissociates slowly and thus likely over-exaggerates client proximity within condensates. The dissociation of the second tool, Dronpa, is more controllable and should be a more accurate measure. Unfortunately, this tool was only used in the last figure. Moreover, it would be important to further validate their findings using another method (e.g. FCS) or adjust conclusions to reflect these limitations

Response: We agree that the Dronpa-based tool provides a more controllable and accurate measure than the Homo-FC tool because of its reversibility. Therefore and also as the reviewer concerned, we additionally applied the Dronpa sensor to the cellular assays described in revised Fig. 2 and Fig. 3. Unlike Homo-FC, Dronpa proteins must be turned-off by 488 nm for 10 sec, and concentration/proximity-dependent tetramerization turn-on signals were subsequently imaged for the next 4 min incubation as described in revised Fig. 5.

First, Dronpa was fused to length-varied mCh-FUS (mCh-2.0 FUS-Dronpa and mCh-0.5 N FUS-Dronpa) and expressed in HeLa cells. As observed with mCh-FUS-Homo-FC constructs (revised Fig. 2), Dronpa with longer 2.0 FUS formed clear cellular condensates, while mCh-0.5 N FUS-Dronpa did not show any noticeable puncta (revised Fig. S20a). Moreover, after Dronpa turn-off by 488 nm light, strong green signals appeared rapidly only by mCh-2.0 FUS-Dronpa (particularly in puncta) but not by mCh-0.5 N FUS-Dronpa, consistent with the Homo-FC experiments.

Next, 0.5N FUS-fused Dronpa (0.5N FUS-Dronpa as client) was co-expressed with mCh-2.0 FUS (as scaffold) to examine Dronpa proximity enhancement inside FUS condensates, as conducted with Homo-FC in revised Fig 3. Again, after Dronpa turn-off by 488 nm light, strong green signals appeared rapidly particularly inside mCh-2.0 FUS condensates (revised Fig. S20b). Overall GFP/mCh values were comparable to those of the Homo-FC probe.

We revised the manuscript with these added experimental data with the Dronpa sensor to

further support observed client proximity enhancement under various cellular client-scaffold conditions. In addition, limitations of applied sensor systems (such as accumulated Homo-FC signals and the need of a turn-off step for Dronpa) were also more clarified with adjusted conclusions, as the reviewer also suggested.

- In Results (Page 9, second paragraph): Multiple sentences “*The Dronpa probe was also applied to FUS-repeat condensates as demonstrated with the Homo-FC probe. When Dronpa was directly fused to length-varied mCh-FUS, Dronpa with longer 2.0 FUS formed clear cellular condensates, while Dronpa with 0.5N FUS did not show any noticeable puncta (Fig S20a). In addition, strong green signals appeared rapidly only from mCh-2.0 FUS-Dronpa (particularly in puncta) but not from mCh-0.5N FUS-Dronpa, consistent with the Homo-FC experiments (Fig. 2). When 0.5N FUS-fused Dronpa (client) was co-expressed with mCh-2.0 FUS (scaffold), again, strong green signals appeared rapidly, particularly inside mCh-2.0 FUS condensates (Fig. S20b).*” were added to demonstrate client proximity enhancement by using the Dronpa probe inside repeated IDP-condensates.

- In Results (Page 8, second paragraph): To state the limitation of the Homo-FC sensor, a sentence was revised to “*Although we observed strong FC enhancement inside cellular condensates, complementation is a unique protein interaction that is nearly irreversible, which might exaggerate client proximity within condensates.*”.

- In Discussion (Page 13, second paragraph): Two sentences “*Still, there are several caveats in our method: FP complementation is nearly irreversible, and thereby, we might observe accumulated signals. In addition, while the Dronpa probe is likely reversible, the mechanism of tetramerization-dependent fluorescent signal generation is not fully understood yet, and a turn-off step by light is necessary.*” were added to further state the limitations of the present tools (Homo-FC and Dronpa) as discussed above.

- In Supplementary Information: A figure (Fig. S20), which shows proximity enhancement of monomerized Dronpa inside repeated-FUS compartments, was added.

- The study mentions one possible functional consequence of enhanced client proximity: increased reaction kinetics. However, the exclusion of non-client molecules may also work similarly. For instance, ‘repelling’ forces between clients and excluded components may be enhanced if the client ‘effective concentration’ is higher in condensates. Another consequence may be in tuning the biophysical properties of the condensates. Testing these ideas would further strengthen the manuscript.

Response: To experimentally evaluate the effects of the exclusion of non-client molecules inside condensates, we directed free Dronpa (without IDR fusion) recruitment inside LAF condensates as discussed in the above second comment. The client proximity enhancement was not observed without IDR fusion on a client even inside condensates (revised Fig. S23). We suppose that the exclusion of non-client molecules has minimal effects on concentration/proximity-dependent tetramerization of Dronpa under our experimental conditions.

As the reviewer also questioned the possible changes of condensate biophysical properties, we examined the diffusivity change of condensates by diverse recruited clients. Interestingly, FRAP recovery profiles clearly indicate that a scaffold mobile fraction was reduced by added client proteins (revised Fig. S25). For example, when 2 μ M LAF-Dronpa (client) was mixed with LAF (scaffold)

condensates, the scaffold mobile fraction was reduced from nearly 90% (no client) to 50%. The mobile fraction of client proteins was also slightly reduced. The mobile fraction reduction increased as the added client concentration increased. In addition, the LAF-Dronpa client (stronger scaffold binding) was more effective for mobile fraction reduction than the 0.5C LAF-Dronpa (weaker scaffold binding) (Fig. S25a). Scaffold and client mobile fractions were also reduced by a LAF client without Dronpa, but in a slightly lesser degree (Fig. S25d). It is not clear yet how small amounts of client proteins influence the scaffold protein diffusivity inside condensates. More studies will be needed to precisely correlate the reduced protein mobility with the client proximity enhancement.

- In Results (Page 12, last paragraph): A paragraph was added to discuss diffusivity changes by clients as discussed above with an added data (Fig. S25).

- In Supplementary Information: A figure (Fig. S25), which shows diffusivity changes of LAF droplets by recruited clients using FRPA, was added.

- Many of the conclusion would require measuring the ratio between client concentration and proximity read-out (as done in Figure 5), rather than the ratio between scaffold and client.

Response: We agree that cellular client concentrations will be valuable information to better interpret the present client proximity enhancement process. To measure client expression levels, a client must be additionally labeled with a fluorescent protein, and three-color quantitative fluorescent measurements were needed: 1) scaffold, 2) client, and 3) GFP-based client proximity sensor (Homo-FC or Dronpa). Red (mCherry) and blue (BFP) proteins are often used with GFP for three-color imaging. However, we had to cover a wide range of (even extremely weak) proximity signals, which were generated by various client proteins under diverse condensate formation conditions. We concerned that a small degree of unavoidable cross-talks between GFP and other color signals (particularly BFPs) could influence weak proximity signals. In addition, we intended to avoid the fusion of two fluorescent proteins to a client, which might also influence client behaviors such as recruitment into condensates and proximity-mediated interactions.

Therefore, here, we measured overall cellular GFP-based proximity read-out signals, and normalized with overall protein expression levels (mCherry-scaffold). GFP expression levels are similar to mCherry expression levels (revised Fig. 3d). The resulting GFP/mCherry indicates relative and normalized client proximities of diverse client-scaffold systems. The primary goal of our cellular study is to compare relative proximity read-out signals of various clients in the absence and presence of client-recruiting condensates in cells.

Consequently, we carefully revised several conclusions to better reflect the resulting data and the limitations of the present measurement methods.

- In Results (Page 6, first paragraph): Two sentences “*Here, 0.5N FUS-Homo-FC concentration was not directly measured since it requires additional FP (e.g. Blue FP) labeling (and three-color imaging), which might perturb accurate FC signal measurements by FP cross-talks. Therefore, we assumed that 0.5N FUS-Homo-FC enrichment inside 2.0 FUS condensates is similar to that of 0.5N FUS-GFP, which was only ca. two-fold (Fig. 3c).*” were added to state our lack of direct [client] measurement in a cell.

- In Results (Page 8, first paragraph): A sentence was revised to “*Again, it is possible that probe complementation inside condensates is significantly more effective than mere expectation from relative concentration consideration.*”.
- In Results (Page 8, last paragraph): Again to notify indirect measurement of client recruitment, a sentence was revised to “*However, actual concentration enrichment of recruited client molecules inside condensates was only marginal as demonstrated with the GFP-fused DDX client (Fig. S11).*”.
- In Discussion (Page 13, second paragraph): A sentence “*Lastly, although client concentrations and condensate enrichment degrees could be accurately measured in vitro, the cellular environment is different from the test tube condition and we should be careful when comparing the in vitro and in cell data.*” was added to further state the limitations of the present measurements as discussed above.
- In Discussion (Page 14, last paragraph): A sentence was revised to “*Strategies to selectively recruit multiple clients and vary (and accurately measure) client/scaffold concentrations would also be helpful.*”.

Minor comments

- Title: Two aspects of client proximity enhancement inside cellular membrane-less compartments – the two aspects would not be immediately clear to readers. Please consider using a more descriptive title.

Response: As the reviewer suggested, we revised the title to “Client proximity enhancement inside cellular membrane-less compartments governed by client-compartment interactions”.

- Manuscript organization: Many important controls and data were tucked away in supplement, making the manuscript difficult to follow and evaluate. For example, please consider moving results such as Figure S1, S3a, S4d, S9b into the main figure.

Response: As the reviewer suggested, we moved Figs S1, S3a, S4d, and S9b in to the main figures, and thereby, the original Fig. 1 was divided to Fig. 1 and Fig. 2.

- A Fig. S1 was moved to Fig. 1a.
- A Fig. S3a was moved to Figs 2cde.
- A Fig. S4d was moved to Fig. 3d.
- A Fig. S9b was moved to Fig. 4f.

- Cell images are often too small to be able to evaluate. Please zoom in on all images and include insets focusing on a few condensates. In addition, please make scale bars clearly visible and consider using greyscale to enhance contrast. In many of the images, the cells often look very rounded, which is unusual for HeLa cells. Please include a measure of cell viability during these experiments.

Response: As the reviewer requested, we zoomed in most cell images and added more visible scale bars.

Regarding rounded cell images, since many of our condensates formed in the nucleus, images were taken in the middle of cells. Therefore, cell images were more in rounded shapes, and the nucleus looked relatively large.

As the reviewer also requested, we measured cell viability during client and scaffold transfection. We transfected HeLa cells with the DDX-Dronpa client and mCh-CRY2olig-FUS (or mCh-CRY2olig-LAF) scaffold proteins. After 20 h incubation (cell imaging condition), cell viability was similar to that of un-transfected cells (Fig. S19c).

- Most cell images were enlarged at least 1.2 fold, and more visible scale bars were added.
- In Results (Page 4, second paragraph): A sentence “*Most FUS condensates were observed in the nucleus.*” was added.
- In Supplementary Information: A figure (Fig. S5), which shows fluorescence (mCh and GFP) images of cells expressing mCh-1.5 FUS or mCh-2.0 FUS (scaffold) and 0.5N FUS-Homo-FC (client) with DAPI images, was added with a note on how some images were obtained with a focus in the middle of cells (focusing around the nucleus).
- In Supplementary Information: A figure (Fig. S19c), which shows cell viabilities of HeLa cells transfected with a mCh-CRY2olig-IDR scaffold and a DDX-Dronpa client at 20 h after transfection, was added.

- Please include n of cells analyzed for all experiments in legend.

Response: As the reviewer requested, n of cells analyzed was added to all experimental legends.

- Many images seem over-exposed (particularly mCh) and GFP appears to not perfectly colocalize with the mCh signal as a result. Please show a line-profile or similar to show whether this is true.

Response: Here, we tried to examine cells with a wide range of protein expression levels, in order to investigate expression level-dependent proximity signal changes. Consequently, as the reviewer noted, some cells show rather strong mCh signals and sometimes saturated signals particularly at condensates.

In order to minimize cross-talks, cells were sequentially scanned from mCh to GFP, and accordingly, there is a time gap of 2-3 seconds between mCh- and GFP-scanned images. There was some condensate movement during this time gap and consequently imperfect co-localization during live cell imaging. In the revised manuscript, we provided line-profiles (both mCh and GFP) of two representative live cell images (revised Figs 4d and 5b). Most mCh and GFP signals were co-localized nicely in these images, while puncta with imperfect co-localization were also observed (Figs S9b and S19d), as the reviewer noted.

- In Results (Page 7, second paragraph): A sentence “*A portion of IDR condensates could also rapidly move inside cells even during a short time gap between sequential mCh and GFP imaging, which caused sporadic imperfect mCh-GFP co-localization (Fig. S9b).*” was added.
- In Supplementary Information: Two figures (Fig. S9b and Fig. 19d), which show fluorescence line-profiles of mCh (scaffold: mCh-CRY2olig-LAF) and GFP (client: DDX-Homo-FC in Fig. S9b or

DDX-Dronpa in Fig. S19d) signals, were added.

- The term ‘effective concentration’ seems confusing and potentially misleading, given that the tools measure the proximity of molecules to each other. Please define this term explicitly or replace it with proximity.

Response: As the reviewer noted and also discussed in the above second major comment, the term ‘effective concentration’ was replaced with ‘overall proximity enhancement factor’.

- In Abstract: A sentence was revised to “*By employing an in vitro phase separation model, we discovered that the operational proximity of clients (measured from client-client interactions) could be over 16 times higher than the expected proximity from actual client concentrations inside compartments.*”.

- In Introduction (Page 3, third paragraph): A sentence was revised to “*In vitro experiments indicated that the acting proximity of clients inside condensates is significantly higher than the expected proximity based on actual inner client concentrations.*”.

- In Results (Page 10-12): The section “**Quantitative measurement of FP interaction enhancement inside IDR compartments**” was thoroughly revised to introduce ‘proximity enhancement factor’ as we heavily discussed in the second comment.

- Please state the sensitivity / proximity range of the FC and Dronpa. This information is crucial to assess how direct the interactions reported by the tools are likely to be.

Response: To obtain the sensitivity/proximity ranges of the fluorescent protein sensors, proximity-dependent fluorescence signals of these proteins at varying concentrations must be precisely measured, likely in a well-defined in vitro condition. However, Homo-FC constructs are fairly unstable outside cells and prone to aggregation particularly at a high concentration, which restricted accurate proximity-dependent signal measurement. On the other hand, we were able to measure Dronpa tetramerization turn-on signals as a function of Dronpa concentration as shown in our in vitro assays. Proximity signals of LAF-Dronpa (as well as 0.5C LAF-Dronpa) were reliably measured from 1 μM and up to 50 μM at a fixed imaging condition (i.e. excitation laser power, expose time). Sensitivity-wise, the Dronpa signals at 1 μM was over two-fold higher than that at 0 μM , indicating 1 μM as the limit of detection (see Source Data). However, the proximity range (up to 50 μM) could not be further extended to higher [LAF-Dronpa], since LAF-Dronpa became rather unstable at higher concentrations. As the reviewer noted, this information is important to assess the present in vitro assay system, and thereby, we revised the manuscript to contain this information on the sensitivity/proximity range of the Dronpa sensor.

- In Results (Page 10, first paragraph): A sentence “*LAF-Homo-FC was prone to aggregation, and therefore, could not be used for this quantitative in vitro measurements.*” was added.

- In Supplementary Information (Fig. S22): A paragraph was added as a Note to discuss the sensitivity and range of the Dronpa client probe in vitro.

- Please avoid vague statements that appear throughout the manuscript. E.g. “how specific clients are selectively enriched and distinctly react”; “biomolecular behaviors inside these liquid compartments have been largely unexplored.”; “understanding of biomolecular behaviors inside membrane-less compartments”; Many but not all “Membrane-less compartments show remarkable liquid-like properties”

Response: As the reviewer requested, listed statements were removed or revised.

- In Abstract: A sentence was revised to “*Although their formation mechanisms have been steadily elucidated via the classical concept of liquid-liquid phase separation, biomolecular behaviors such as protein interactions inside these liquid compartments have been largely unexplored.*”.

- In Introduction (Page 2, first paragraph): A sentence was revised to “*Many membrane-less compartments show remarkable liquid-like properties...*”.

- In Introduction (Page 2, second paragraph): A sentence was revised to “*...our understanding of biomolecular behaviors such as protein interactions and enzymatic reactions inside membrane-less compartments is still very limited.*”.

- In Introduction (Page 3, second paragraph): A sentence was revised to “*To fully elucidate the working principles of these compartments as temporal reaction centers in cells, it is important to understand how selectively enriched clients differently react inside protein compartments compared to outside.*”.

- Correct “multivalent interactions between repeated proteins” on page 2 to “repeated folded protein domains”.

Response: The above phrase was corrected as suggested.

- To support the statements on page 2, the following reference seems very relevant. Kim, Tsang...Forman-Kay 2019 demonstrated translational control by differential partitioning of components.

Response: As the reviewer suggested, the above reference was added.

- In Introduction (Page 2, second paragraph): A sentence “*An RNA deadenylation rate was also enhanced in in vitro compartments formed with IDRs of two interacting translation regulating proteins and RNA.¹⁶*” was added with the above reference.

- Based on images in 2C, it seems that a significant portion of the nucleus is bleached for FRAP measurements, not the individual condensates. This may be why the recovery times are significantly slower compared to what has been reported for FUS.

Response: As the reviewer noted, the overall mCh signals in the nucleus was reduced by

photobleaching. To compensate this un-intended bleaching, we measured mean fluorescent intensity ratios between the region of bleaching (ROB) and the total nucleus (I_{ROB}/I_{TOT}) in the case of Fig. 1c. We normalized that I_{ROB}/I_{TOT} equals 1 before bleach and 0 after bleach. We believe that this normalization can minimize biases from whole condensate bleaching. We revised the method to provide better description of the present FRAP analysis.

- In Methods (Page 15): The FRAP method was revised as discussed above.

- Please include DAPI to orient us in 2E.

Response: As the reviewer requested, we added a cellular DAPI image of the revised Fig. 3f experiment (revised Fig. S5). Again, as we discussed above, since images were obtained with a focus in the middle of cells (focusing around the nucleus), the size of the nucleus was rather large.

- In Supplementary Information: A figure (Fig. S5), which shows Fluorescence (mCh and GFP) images of cells expressing mCh-1.5 FUS or mCh-2.0 FUS (scaffold) and 0.5N FUS-Homo-FC (client) with DAPI images, was added.

Reviewer 2

General remarks and comments

- This is an interesting paper with an extensive amount of work for making condensates with a variety of different molecular tools, and correspondingly measuring or inferring the degree of concentration of clients in the scaffolds. I will not summarize all the molecules and clients that were used here, although I would say the presentation of the paper would be greatly included if the authors themselves provided a table of LLPS forming molecules, clients, and a summary of the keep result with this combination.

Response: As the reviewer requested, we added a summary table of LLPS forming scaffold/client proteins and proximity (GFP/mCh) signals (Table S1)

- The point of the paper that the authors most want to make is that the effective interaction between clients is enhanced by their proximity in the droplets beyond their simple concentration. This would imply that trapping the clients in a local volume would enhance their interaction. A simple energy model to this effect is provided, and suggested that for molecules with a binding one can deduce the effective constrained volume.

- There are two systems used to describe this effective increase in client interactions. One, which is less convincing, involves homoFC constructs. The other, which is more convincing, involves the recruitment of LAF-Dronpa-Cy5 clients to LAF droplets, The cy5 measures the actual concentration

while the Dronpa tetramerization measures the enhancement effect, Although I question whether the enhancement is due to the tetrameric interactions (versus the the dimerization required for homoFc), there is a clear indication of enhancement, as represented in Figure 5.

Response: As the reviewer noted, Homo-FC has a clear limitation of irreversibility, which could yield accumulated signals, while Dronpa could be more controllable and accurate. However, it is not clear yet how the tetramerization process (rather than dimerization) during Dronpa turn-on affects observed proximity enhancement. Nonetheless, we experimentally determined overall proximity enhancement factors in the revised manuscript. The potential effect of the tetrameric interactions of Dronpa was also discussed in the revised manuscript.

- In Results (Page 8, second paragraph): To state the limitation of the Homo-FC sensor, a sentence was revised to *“Although we observed strong FC enhancement inside cellular condensates, complementation is a unique protein interaction that is nearly irreversible, which might exaggerate client proximity within condensates.”*

- In Discussion (Page 13, second paragraph): Two sentences *“Still, there are several caveats in our method: FP complementation is nearly irreversible, and thereby, we might observe accumulated signals. In addition, while the Dronpa probe is likely reversible, the mechanism of tetramerization-dependent fluorescent signal generation is not fully understood yet, and a turn-off step by light is necessary.”* were added as discussed above.

• Other experiments presented here do not address the enhancement, but do add to the impressive ability of this group to engineer phase separation across a wide range of current protein chemistries. Overall, this paper is an impressive tour de force and I favor publication.

Reviewer 3

General remarks

This is an interesting and very important study that adds significantly to the field and will have a noticeable impact. The manuscript is well-written and concise. It reports impressive results of a set of well-thought and masterly performed experiments that shed light on the molecular mechanisms of biological liquid-liquid phase transitions (LLPTs) and formation of membrane-less organelle (MLOs). The authors showed that the operational effective concentrations inside the MLOs (i.e. concentrations defining interactions between the clients and based on the measured client-client interactions in MLOs) could significantly exceed the actual concentrations of the compounds of MLOs. These effect was defined as the client proximity enhancement by the MLOs. The authors also proposed that such client proximity enhancement can be defined by a least two mechanisms, such as selective recruitment of the specific clients into MLOs and specific co-localization of clients (or MLO passenger proteins) in close proximity to the MLO-forming scaffold proteins (or LLPT drivers), leading to further increase in local client concentrations. Obviously, these observations re crucial for better understanding of basic principles driving biological LLPTs and related to the MLO biogenesis. However, there are two issues

that need to be addressed.

- The authors briefly mentioned in introduction section that recent studies indicated that “multivalent interactions between repeated proteins or intrinsically disordered proteins/regions (IDPs/IDRs) [4] can drive liquid-liquid phase separation (LLPS), and that this is the major formation principle of compartmentalized biomolecular condensates (also termed droplets) [5].” The provided references to two publications here, 4: Dyson, H.J. & Wright, P.E. Intrinsically unstructured proteins and their functions. *Nat. Rev. Mol. Cell Biol.* 6, 197-208 (2005); and 5: Alberti, S., Gladfelter, A. & Mittag, T. Considerations and Challenges in Studying Liquid-Liquid Phase Separation and Biomolecular Condensates. *Cell* 176, 419-434 (2019). Although reference [4] represents one of the important reviews on IDPs, it does not discuss the roles of multivalent interactions between IDPs/IDRs in LLPS. However, there are several focused studies, where the role of such interactions between IDPs/IDRs are discussed. Furthermore, there are also several focused studies on the overall abundance of IDPs/IDRs in MLOs. In my view, these studies should be cited and discussed in this manuscript.

Response: As the reviewer suggested, several references (e.g. *Curr Opin Struct Biol* 2017, 44, 18-30) were added to this sentence, and references were re-organized to better support the sentence.

- In Introduction (Page 2, first paragraph): A sentence “*Recent studies indicate that repeated folded protein domains⁴ or intrinsically disordered proteins/regions (IDPs/IDRs)⁵⁻⁷ can drive liquid-liquid phase separation (LLPS), and that this is the major formation principle of compartmentalized biomolecular condensates (also termed droplets)^{2, 8, 9}.*” was added with the reorganized and added references.

- Although the authors mentioned that there is a selective recruitment of the specific clients/passengers into MLOs, they did not discuss mechanisms of such specific recruitment. However, there are several recent studies indicating that such recruitment can be a reflection of specific partitioning of clients/passengers into MLOs; i.e., a process similar to the partitioning of various substances into different phases of aqueous two-phase systems (ATPSs). In my view, these studies should be cited and discussed in this manuscript.

Response: As the reviewer requested, we added multiple recent references, where specific partitioning of clients/passengers into MLOs and resulting MLO activity changes were studied.

- In Introduction (Page 2-3): Two sentences “*Recent studies indicated that specific partitioning of clients and scaffolds in condensates can be influenced by various factors such as crowding environments²⁰ and condensate compositions²¹. Dynamic mRNA partitioning to cellular stress granules or to processing bodies was also reported²², and a theoretical model was used to study how client-scaffold interactions govern condensate stability²³.*” were added with new references.

REVIEWERS' COMMENTS

Reviewer #2 (Remarks to the Author):

The authors addressed my minor comments, and I am happy with the revision.